# General Munchausen Reinforcement Learning with Tsallis Kullback-Leibler Divergence

**Lingwei Zhu**
University of Alberta
`lingwei4@ualberta.ca`

**Zheng Chen**
Osaka University
`chenz@sanken.osaka-u.ac.jp`

**Matthew Schlegel**
University of Alberta
`mkschleg@ualberta.ca`

**Martha White**
University of Alberta
CIFAR Canada AI Chair, Amii
`whitem@ualberta.ca`

## Abstract

Many policy optimization approaches in reinforcement learning incorporate a Kullback-Leilbler (KL) divergence to the previous policy, to prevent the policy from changing too quickly. This idea was initially proposed in a seminal paper on Conservative Policy Iteration, with approximations given by algorithms like TRPO and Munchausen Value Iteration (MVI). We continue this line of work by investigating a generalized KL divergence—called the Tsallis KL divergence. Tsallis KL defined by the $q$-logarithm is a strict generalization, as $q = 1$ corresponds to the standard KL divergence; $q > 1$ provides a range of new options. We characterize the types of policies learned under the Tsallis KL, and motivate when $q > 1$ could be beneficial. To obtain a practical algorithm that incorporates Tsallis KL regularization, we extend MVI, which is one of the simplest approaches to incorporate KL regularization. We show that this generalized MVI($q$) obtains significant improvements over the standard MVI($q = 1$) across 35 Atari games.

## 1 Introduction

There is ample theoretical evidence that it is useful to incorporate KL regularization into policy optimization in reinforcement learning. The most basic approach is to regularize towards a uniform policy, resulting in entropy regularization. More effective, however, is to regularize towards the previous policy. By choosing KL regularization between consecutively updated policies, the optimal policy becomes a softmax over a uniform average of the full history of action value estimates [Vieillard et al., 2020a]. This averaging smooths out noise, allowing for better theoretical results [Azar et al., 2012, Kozuno et al., 2019, Vieillard et al., 2020a, Kozuno et al., 2022].

Despite these theoretical benefits, there are some issues with using KL regularization in practice. It is well-known that the uniform average is susceptible to outliers; this issue is inherent to KL divergence [Futami et al., 2018]. In practice, heuristics such as assigning vanishing regularization coefficients to some estimates have been implemented widely to increase robustness and accelerate learning [Grau-Moya et al., 2019, Haarnoja et al., 2018, Kitamura et al., 2021]. However, theoretical guarantees no longer hold for those heuristics [Vieillard et al., 2020a, Kozuno et al., 2022]. A natural question is what alternatives we can consider to this KL divergence regularization, that allows us to overcome some of these disadvantages while maintaining the benefits associate with restricting aggressive policy changes and smoothing errors.

37th Conference on Neural Information Processing Systems (NeurIPS 2023).

In this work, we explore one possible direction by generalizing to Tsallis KL divergences. Tsallis KL divergences were introduced for physics [Tsallis, 1988, 2009] using a simple idea: replacing the use of the logarithm with the deformed $q$-logarithm. The implications for policy optimization, however, are that we get quite a different form for the resulting policy. Tsallis *entropy* with $q = 2$ has actually already been considered for policy optimization [Chow et al., 2018, Lee et al., 2018], by replacing Shannon entropy with Tsallis entropy to maintain stochasticity in the policy. The resulting policies are called *sparsemax* policies, because they concentrate the probability on higher-valued actions and truncate the probability to zero for lower-valued actions. Intuitively, this should have the benefit of maintaining stochasticity, but only amongst the most promising actions, unlike the Boltzmann policy which maintains nonzero probability on all actions. Unfortunately, using only Tsallis entropy did not provide significant benefits, and in fact often performed worse than existing methods. We find, however, that using a Tsallis KL divergence to the previous policy does provide notable gains.

We first show how to incorporate Tsallis KL regularization into the standard value iteration updates, and prove that we maintain convergence under this generalization from KL regularization to Tsallis KL regularization. We then characterize the types of policies learned under Tsallis KL, highlighting that there is now a more complex relationship to past action-values than a simple uniform average. We then show how to extend Munchausen Value Iteration (MVI) [Vieillard et al., 2020b], to use Tsallis KL regularization, which we call MVI($q$). We use this naming convention to highlight that this is a strict generalization of MVI: by setting $q = 1$, we exactly recover MVI. We then compare MVI($q = 2$) with MVI (namely the standard choice where $q = 1$), and find that we obtain significant performance improvements in Atari.

**Remark:** There is a growing body of literature studying generalizations of KL divergence in RL [Nachum et al., 2019, Zhang et al., 2020]. Futami et al. [2018] discussed the inherent drawback of KL divergence in generative modeling and proposed to use $\beta$- and $\gamma$-divergence to allow for weighted average of sample contribution. These divergences fall under the category known as the $f$-divergence [Sason and Verdú, 2016], commonly used in other machine learning domains including generative modeling [Nowozin et al., 2016, Wan et al., 2020, Yu et al., 2020] and imitation learning [Ghasemipour et al., 2019, Ke et al., 2019]. In RL, Wang et al. [2018] discussed using tail adaptive $f$-divergence to enforce the mass-covering property. Belousov and Peters [2019] discussed the use of $\alpha$-divergence. Tsallis KL divergence, however, has not yet been studied in RL.

## 2 Problem Setting

We focus on discrete-time discounted Markov Decision Processes (MDPs) expressed by the tuple $(\mathcal{S}, \mathcal{A}, d, P, r, \gamma)$, where $\mathcal{S}$ and $\mathcal{A}$ denote state space and finite action space, respectively. Let $\Delta(\mathcal{X})$ denote the set of probability distributions over $\mathcal{X}$. $d \in \Delta(\mathcal{S})$ denotes the initial state distribution. $P : \mathcal{S} \times \mathcal{A} \to \Delta(\mathcal{S})$ denotes the transition probability function, and $r(s, a)$ defines the reward associated with that transition. $\gamma \in (0, 1)$ is the discount factor. A policy $\pi : \mathcal{S} \to \Delta(\mathcal{A})$ is a mapping from the state space to distributions over actions. We define the action value function following policy $\pi$ and starting from $s_0 \sim d(\cdot)$ with action $a_0$ taken as $Q_\pi(s, a) = \mathbb{E}_\pi \left[ \sum_{t=0}^\infty \gamma^t r_t | s_0 = s, a_0 = a \right]$. A standard approach to find the optimal value function $Q_*$ is value iteration. To define the formulas for value iteration, it will be convenient to write the action value function as a matrix $Q_\pi \in \mathbb{R}^{|\mathcal{S}| \times |\mathcal{A}|}$. For notational convenience, we define the inner product for any two functions $F_1, F_2 \in \mathbb{R}^{|\mathcal{S}| \times |\mathcal{A}|}$ over actions as $\langle F_1, F_2 \rangle \in \mathbb{R}^{|\mathcal{S}|}$.

We are interested in the entropy-regularized MDPs where the recursion is augmented with $\Omega(\pi)$:

$$\begin{cases} \pi_{k+1} = \arg\max_\pi \left( \langle \pi, Q_k \rangle - \tau \Omega(\pi) \right), \\ Q_{k+1} = r + \gamma P(\langle \pi_{k+1}, Q_k \rangle - \tau \Omega(\pi_{k+1})) \end{cases} \tag{1}$$

This modified recursion is guaranteed to converge if $\Omega$ is concave in $\pi$. For standard (Shannon) entropy regularization, we use $\Omega(\pi) = -\mathcal{H}(\pi) = \langle \pi, \ln \pi \rangle$. The resulting optimal policy has $\pi_{k+1} \propto \exp\left( \tau^{-1} Q_k \right)$, where $\propto$ indicates *proportional to* up to a constant not depending on actions.

More generally, we can consider a broad class of regularizers known as $f$-divergences [Sason and Verdú, 2016]: $\Omega(\pi) = D_f(\pi \| \mu) := \langle \mu, f(\pi/\mu) \rangle$, where $f$ is a convex function. For example, the KL divergence $D_{KL}(\pi \| \mu) = \langle \pi, \ln \pi - \ln \mu \rangle$ can be recovered by $f(t) = -\ln t$. In this work, when we say KL regularization, we mean the standard choice of setting $\mu = \pi_k$, the estimate from the previous update. Therefore, $D_{\text{KL}}$ serves as a penalty to penalize aggressive policy changes. The

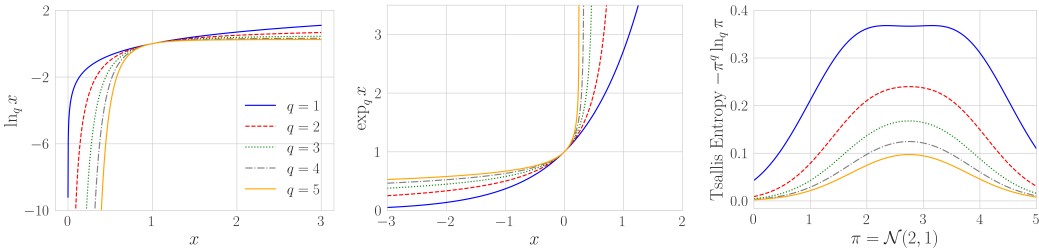

Figure 1: $\ln_q x$, $\exp_q x$ and Tsallis entropy component $-\pi^q \ln_q \pi$ for $q = 1$ to $5$. When $q = 1$ they respectively recover their standard counterpart. $\pi$ is chosen to be Gaussian $\mathcal{N}(2, 1)$. As $q$ gets larger $\ln_q x$ (and hence Tsallis entropy) becomes more flat and $\exp_q x$ more steep.

optimal policy in this case takes the form $\pi_{k+1} \propto \pi_k \exp\left(\tau^{-1} Q_k\right)$. By induction, we can show this KL-regularized optimal policy $\pi_{k+1}$ is a softmax over a uniform average over the history of action value estimates [Vieillard et al., 2020a]: $\pi_{k+1} \propto \pi_k \exp\left(\tau^{-1} Q_k\right) \propto \cdots \propto \exp\left(\tau^{-1} \sum_{j=1}^{k} Q_j\right)$. Using KL regularization has been shown to be theoretically superior to entropy regularization in terms of error tolerance [Azar et al., 2012, Vieillard et al., 2020a, Kozuno et al., 2022, Chan et al., 2022].

The definitions of $\mathcal{H}(\cdot)$ and $D_{\mathrm{KL}}(\cdot||\cdot)$ rely on the standard logarithm and both induce softmax policies as an exponential (inverse function) over (weighted) action-values [Hiriart-Urruty and Lemaréchal, 2004, Nachum and Dai, 2020]. Convergence properties of the resulting regularized algorithms have been well studied [Kozuno et al., 2019, Geist et al., 2019, Vieillard et al., 2020a]. In this paper, we investigate Tsallis entropy and Tsallis KL divergence as the regularizer, which generalize Shannon entropy and KL divergence respectively.

## 3 Generalizing to Tsallis Regularization

We can easily incorporate other regularizers in to the value iteration recursion, and maintain convergence as long as those regularizers are strongly convex in $\pi$. We characterize the types of policies that arise from using this regularizer, and prove the convergence of resulting regularized recursion.

### 3.1 Tsallis Entropy Regularization

Tsallis entropy was first proposed by Tsallis [1988] and is defined by the $q$-logarithm. The $q$-logarithm and its unique inverse function, the $q$-exponential, are defined as:

$$\ln_q x := \frac{x^{1-q} - 1}{1 - q}, \quad \exp_q x := [1 + (1 - q)x]_+^{\frac{1}{1-q}}, \quad \text{for } q \in \mathbb{R} \backslash \{1\} \tag{2}$$

where $[\cdot]_+ := \max\{\cdot, 0\}$. We define $\ln_1 = \ln$, $\exp_1 = \exp$, as in the limit $q \to 1$, the formulas in Eq. (2) approach these functions. Tsallis entropy can be defined by $S_q(\pi) := p\langle -\pi^q, \ln_q \pi\rangle$, $p \in \mathbb{R}$ [Suyari and Tsukada, 2005]. We visualize the $q$-logarithm, $q$-exponential and Tsallis entropy for different $q$ in Figure 1. As $q$ gets larger, $q$-logarithm (and hence Tsallis entropy) becomes more flat and $q$-exponential more steep[1]. Note that $\exp_q$ is only invertible for $x > \frac{-1}{1-q}$.

Tsallis policies have a similar form to softmax, but using the $q$-exponential instead. Let us provide some intuition for these policies. When $p = \frac{1}{2}$, $q = 2$, $S_2(\pi) = \frac{1}{2}\langle \pi, 1 - \pi\rangle$, the optimization problem $\arg\max_{\pi \in \Delta(\mathcal{A})} \langle \pi, Q\rangle + S_2(\pi) = \arg\min_{\pi \in \Delta(\mathcal{A})} \|\pi - Q\|_2^2$ is known to be the Euclidean projection onto the probability simplex. Its solution $[Q - \psi]_+$ is called the sparsemax [Martins and Astudillo, 2016, Lee et al., 2018] and has sparse support [Duchi et al., 2008, Condat, 2016, Blondel et al., 2020]. $\psi : \mathcal{S} \times \mathcal{A} \to \mathcal{S}$ is the unique function satisfying $\langle \mathbf{1}, [Q - \psi]_+\rangle = 1$.

As our first result, we unify the Tsallis entropy regularized policies for all $q \in \mathbb{R}_+$ with the $q$-exponential, and show that $q$ and $\tau$ are interchangeable for controlling the truncation.

---

[1] The $q$-logarithm defined here is consistent with the physics literature and different from prior RL works [Lee et al., 2020], where a change of variable $q^* = 2 - q$ is made. We analyze both cases in Appendix A.

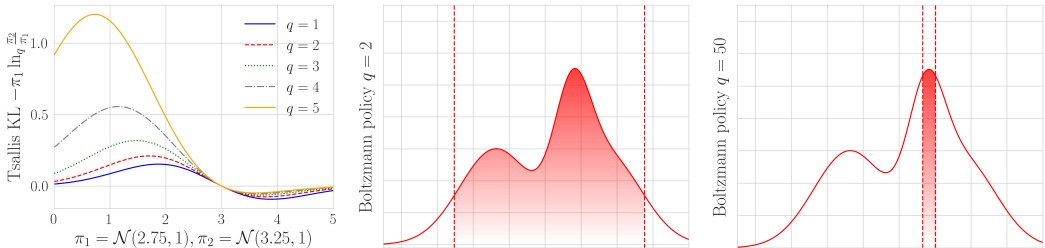

Figure 2: (Left) Tsallis KL component $-\pi_1 \ln_q \frac{\pi_2}{\pi_1}$ between two Gaussian policies $\pi_1 = \mathcal{N}(2.75, 1), \pi_2 = \mathcal{N}(3.25, 1)$ for $q = 1$ to 5. When $q = 1$ TKL recovers KL. For $q > 1$, TKL is more mode-covering than KL. (Mid) The sparsemax operator acting on a Boltzmann policy when $q = 2$. (Right) The sparsemax when $q = 50$. Truncation gets stronger as $q$ gets larger. The same effect can be also controlled by $\tau$.

**Theorem 1.** *Let $\Omega(\pi) = -S_q(\pi)$ in Eq. (1). Then the regularized optimal policies can be expressed:*

$$\pi(a|s) = \sqrt[1-q]{\left[\frac{Q(s,a)}{\tau} - \tilde{\psi}_q\left(\frac{Q(s,\cdot)}{\tau}\right)\right]_+ (1-q)} = \exp_q\left(\frac{Q(s,a)}{\tau} - \psi_q\left(\frac{Q(s,\cdot)}{\tau}\right)\right) \quad (3)$$

*where $\psi_q = \tilde{\psi}_q + \frac{1}{1-q}$. Additionally, for an arbitrary $(q, \tau)$ pair with $q > 1$, the same truncation effect (support) can be achieved using $(q = 2, \frac{\tau}{q-1})$.*

*Proof.* See Appendix B for the full proof. $\qquad \square$

Theorem 1 characterizes the role played by $q$: controlling the degree of truncation. We show the truncation effect when $q = 2$ and $q = 50$ in Figure 2, confirming that Tsallis policies tend to truncate more as $q$ gets larger. The theorem also highlights that we can set $q = 2$ and still get more or less truncation using different $\tau$, helping to explain why in our experiments $q = 2$ is a generally effective choice.

Unfortunately, the threshold $\tilde{\psi}_q$ (and $\psi_q$) does not have a closed-form solution for $q \neq 1, 2, \infty$. Note that $q = 1$ corresponds to Shannon entropy and $q = \infty$ to no regularization. However, we can resort to Taylor's expansion to obtain *approximate sparsemax policies*.

**Theorem 2.** *For $q \neq 1, \infty$, we can obtain approximate threshold $\hat{\psi}_q \approx \psi_q$ using Taylor's expansion, and therefore an approximate policy:*

$$\hat{\pi}(a|s) \propto \exp_q\left(\frac{Q(s,a)}{\tau} - \hat{\psi}_q\left(\frac{Q(s,\cdot)}{\tau}\right)\right), \quad \hat{\psi}_q\left(\frac{Q(s,\cdot)}{\tau}\right) \doteq \frac{\sum_{a \in K(s)} \frac{Q(s,\cdot)}{\tau} - 1}{|K(s)|} + 1. \quad (4)$$

$K(s)$ *is the set of highest-valued actions, satisfying the relation* $1 + i\frac{Q(s,a_{(i)})}{\tau} > \sum_{j=1}^{i} \frac{Q(s,a_{(j)})}{\tau}$, *where $a_{(j)}$ indicates the action with $j$th largest action value. The sparsemax policy sets the probabilities of lowest-valued actions to zero:* $\pi(a_{(i)}|s) = 0, i = z+1, \ldots, |\mathcal{A}|$ *where* $Q(s, a_{(z)}) > \tau^{-1}\hat{\psi}_q\left(\frac{Q(s,\cdot)}{\tau}\right) > Q(s, a_{(z+1)})$. *When $q = 2$, $\hat{\psi}_q$ recovers $\psi_q$.*

*Proof.* See Appendix B for the full proof. $\qquad \square$

Lee et al. [2020] also used $\exp_q$ to represent policies but they consider the continuous action setting and do not give any computable threshold. By contrast, Theorem 2 presents an easily computable $\hat{\psi}_q$ for all $q \notin \{1, \infty\}$.

### 3.2 Tsallis KL Regularization and Convergence Results

The Tsallis KL divergence is defined as $D_{KL}^q(\pi \| \mu) := \langle \pi, -\ln_q \frac{\mu}{\pi} \rangle$ [Furuichi et al., 2004]. It is a member of $f$-divergence and can be recovered by choosing $f(t) = -\ln_q t$. As a divergence penalty,

it is required that $q > 0$ since $f(t)$ should be convex. We further assume that $q > 1$ to align with standard divergences; i.e. penalize large value of $\frac{\pi}{\mu}$, since for $0 < q < 1$ the regularization would penalize $\frac{\mu}{\pi}$ instead. In practice, we find that $0 < q < 1$ tend to perform poorly. In contrast to KL, Tsallis KL is more *mass-covering*; i.e. its value is proportional to the $q$-th power of the ratio $\frac{\pi}{\mu}$. When $q$ is big, large values of $\frac{\pi}{\mu}$ are strongly penalized [Wang et al., 2018]. This behavior of Tsallis KL divergence can also be found in other well-known divergences: the $\alpha$-divergence [Wang et al., 2018, Belousov and Peters, 2019] coincides with Tsallis KL when $\alpha = 2$; Rényi's divergence also penalizes large policy ratio by raising it to the power $q$, but inside the logarithm, which is therefore an additive extension of KL [Li and Turner, 2016]. In the limit of $q \to 1$, Tsallis entropy recovers Shannon entropy and the Tsallis KL divergence recovers the KL divergence. We plot the Tsallis KL divergence behavior in Figure 2.

Now let us turn to formalizing when value iteration under Tsallis regularization converges. The $q$-logarithm has the following properties: *Convexity:* $\ln_q \pi$ is convex for $q \le 0$, concave for $q > 0$. When $q = 0$, both $\ln_q, \exp_q$ become linear. *Monotonicity:* $\ln_q \pi$ is monotonically increasing with respect to $\pi$. These two properties can be simply verified by checking the first and second order derivative. We prove in Appendix A the following similarity between Shannon entropy (reps. KL) and Tsallis entropy (resp. Tsallis KL). *Bounded entropy*: we have $0 \le \mathcal{H}(\pi) \le \ln|\mathcal{A}|$; and $\forall q, 0 \le S_q(\pi) \le \ln_q|\mathcal{A}|$. *Generalized KL property*: $\forall q, D_{KL}^q(\pi \,\|\, \mu) \ge 0$. $D_{KL}^q(\pi \,\|\, \mu) = 0$ if and only if $\pi = \mu$ almost everywhere, and $D_{KL}^q(\pi \,\|\, \mu) \to \infty$ whenever $\pi(a|s) > 0$ and $\mu(a|s) = 0$.

However, despite their similarity, a crucial difference is that $\ln_q$ is non-extensive, which means it is not additive [Tsallis, 1988]. In fact, $\ln_q$ is only *pseudo-additive*:

$$\ln_q \pi\mu = \ln_q \pi + \ln_q \mu + (1 - q)\ln_q \pi \ln_q \mu. \tag{5}$$

Pseudo-additivity complicates obtaining convergence results for Eq. (1) with $q$-logarithm regularizers, since the techniques used for Shannon entropy and KL divergence are generally not applicable to their $\ln_q$ counterparts. Moreover, deriving the optimal policy may be nontrivial. Convergence results have only been established for Tsallis entropy [Lee et al., 2018, Chow et al., 2018].

We know that Eq. (1) with $\Omega(\pi) = D_{KL}^q(\pi \,\|\, \mu)$, for any $\mu$, converges for $q$ that make $D_{KL}^q(\pi \,\|\, \mu)$ strictly convex [Geist et al., 2019]. When $q = 2$, it is strongly convex, and so also strictly convex, guaranteeing convergence.

**Theorem 3.** *The regularized recursion Eq. (1) with $\Omega(\pi) = D_{KL}^q(\pi \,\|\, \cdot)$ when $q = 2$ converges to the unique regularized optimal policy.*

*Proof.* See Appendix C. It simply involves proving that this regularizer is strongly convex. $\square$

### 3.3 TKL Regularized Policies Do More Than Averaging

We next show that the optimal regularized policy under Tsallis KL regularization does more than uniform averaging. It can be seen as performing a weighted average where the degree of weighting is controlled by $q$. Consider the recursion

$$\begin{cases} \pi_{k+1} = \arg\max_\pi \langle \pi, Q_k - D_{KL}^q(\pi \,\|\, \pi_k) \rangle, \\ Q_{k+1} = r + \gamma P \langle \pi_{k+1}, Q_k - D_{KL}^q(\pi_{k+1} || \pi_k) \rangle, \end{cases} \tag{6}$$

where we dropped the regularization coefficient $\tau$ for convenience.

**Theorem 4.** *The greedy policy $\pi_{k+1}$ in Equation (6) satisfies*

$$\pi_{k+1} \propto \left(\exp_q Q_1 \cdots \exp_q Q_k\right) = \left[\exp_q\left(\sum_{j=1}^k Q_j\right)^{q-1} + \sum_{j=2}^k (q-1)^j \sum_{i_1 = 1 < \cdots < i_j}^k Q_{i_1} \cdots Q_{i_j}\right]^{\frac{1}{q-1}}. \tag{7}$$

*When $q = 1$, Eq. (6) reduces to KL regularized recursion and hence Eq. (7) reduces to the KL-regularized policy. When $q = 2$, Eq. (7) becomes:*

$$\exp_2 Q_1 \cdots \exp_2 Q_k = \exp_2\left(\sum_{j=1}^k Q_j\right) + \sum_{\substack{j=2 \\ i_1 = 1 < \cdots < i_j}}^k Q_{i_1} \cdots Q_{i_j}.$$

*i.e., Tsallis KL regularized policies average over the history of value estimates as well as computing the interaction between them* $\sum_{j=2}^{k} \sum_{i_1 < \cdots < i_j}^{k} Q_{i_1} \ldots Q_{i_j}$.

*Proof.* See Appendix D for the full proof. The proof comprises two parts: the first part shows $\pi_{k+1} \propto \exp_q Q_1 \ldots \exp_q Q_k$, and the second part establishes the *more-than-averaging* property by two-point equation [Yamano, 2002] and the $2 - q$ duality [Naudts, 2002, Suyari and Tsukada, 2005] to conclude $\left( \exp_q x \cdot \exp_q y \right)^{q-1} = \exp_q (x + y)^{q-1} + (q - 1)^2 xy$.  $\square$

The form of this policy is harder to intuit, but we can try to understand each component. The first component actually corresponds to a weighted averaging by the property of the $\exp_q$:

$$\exp_q \left( \sum_{i=1}^{k} Q_i \right) = \exp_q Q_1 \exp_q \left( \frac{Q_2}{1 + (1 - q)Q_1} \right) \ldots \exp_q \left( \frac{Q_k}{1 + (1 - q) \sum_{i=1}^{k-1} Q_i} \right). \quad (8)$$

Eq. (8) is a possible way to expand the summation: the left-hand side of the equation is what one might expect from conventional KL regularization; while the right-hand side shows a weighted scheme such that any estimate $Q_j$ is weighted by the summation of estimates before $Q_j$ times $1 - q$ (Note that we can exchange 1 and $q$, see Appendix A). Weighting down numerator by the sum of components in the demoninator has been analyzed before in the literature of weighted average by robust divergences, e.g., the $\gamma$-divergence [Futami et al., 2018, Table 1]. Therefore, we conjecture this functional form helps weighting down the magnitude of excessively large $Q_k$, which can also be controlled by choosing $q$. In fact, obtaining a weighted average has been an important topic in RL, where many proposed heuristics coincide with weighted averaging [Grau-Moya et al., 2019, Haarnoja et al., 2018, Kitamura et al., 2021].

Now let us consider the second term with $q = 2$, therefore the leading $(q - 1)^j$ vanishes. The action-value cross-product term can be intuitively understood as further increasing the probability for any actions that have had consistently larger values across iterations. This observation agrees with the mode-covering property of Tsallis KL. However, there is no concrete evidence yet how the average inside $q$-exponential and the cross-product action values may work jointly to benefit the policy, and their benefits may depend on the task and environments, requiring further categorization and discussion. Empirically, we find that the nonlinearity of Tsallis KL policies bring superior performance to the uniform averaging KL policies on the testbed considered.

## 4   A Practical Algorithm for Tsallis KL Regularization

In this section we provide a practical algorithm for implementing Tsallis regularization. We first explain why this is not straightforward to simply implement KL-regularized value iteration, and how Munchausen Value Iteration (MVI) overcomes this issue with a clever implicit regularization trick. We then extend this algorithm to $q > 1$ using a similar approach, though now with some approximation due once again to the difficulties of pseudo-additivity.

### 4.1   Implicit Regularization With MVI

Even for the standard KL, it is difficult to implement KL-regularized value iteration with function approximation. The difficulty arises from the fact that we cannot exactly obtain $\pi_{k+1} \propto \pi_k \exp (Q_k)$. This policy might not be representable by our function approximator. For $q = 1$, one needs to store all past $Q_k$ which is computationally infeasible.

An alternative direction has been to construct a different value function iteration scheme, which is equivalent to the original KL regularized value iteration [Azar et al., 2012, Kozuno et al., 2019]. A recent method of this family is Munchausen VI (MVI) [Vieillard et al., 2020b]. MVI implicitly enforces KL regularization using the recursion

$$\begin{cases} \pi_{k+1} = \arg\max_{\pi} \langle \pi, Q_k - \tau \ln \pi \rangle \\ Q_{k+1} = r + \textcolor{red}{\alpha \tau \ln \pi_{k+1}} + \gamma P \langle \pi_{k+1}, Q_k - \textcolor{blue}{\tau \ln \pi_{k+1}} \rangle \end{cases} \quad (9)$$

We see that Eq. (9) is Eq. (1) with $\Omega(\pi) = -\mathcal{H}(\pi)$ (blue) plus an additional red *Munchausen term*, with coefficient $\alpha$. Vieillard et al. [2020b] showed that implicit KL regularization was performed

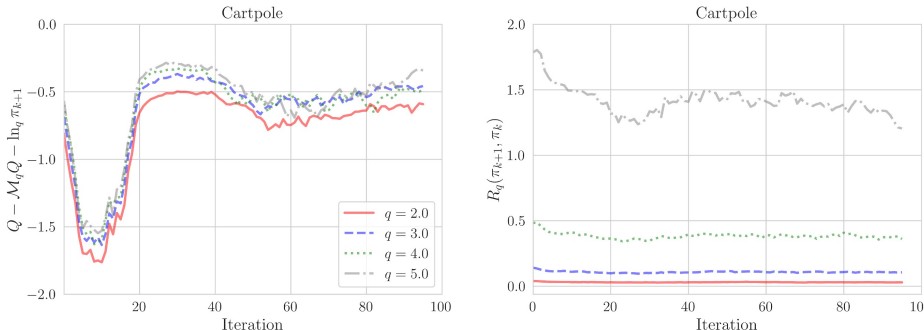

Figure 3: MVI($q$) on CartPole-v1 for $q = 2, 3, 4, 5$, averaged over 50 seeds, with $\tau = 0.03, \alpha = 0.9$. (Left) The difference between the proposed action gap $Q_k - \mathcal{M}_{q,\tau}Q_k$ and the general Munchausen term $\ln_q \pi_{k+1}$ converges to a constant. (Right) The residual $R_q(\pi_{k+1}, \pi_k)$ becomes larger as $q$ increases. For $q = 2$, it remains negligible throughout the learning.

under the hood, even though we still have tractable $\pi_{k+1} \propto \exp\left(\tau^{-1} Q_k\right)$:

$$Q_{k+1} = r + \alpha\tau \ln \pi_{k+1} + \gamma P \langle \pi_{k+1}, Q_k - \tau \ln \pi_{k+1} \rangle \Leftrightarrow Q_{k+1} - \alpha\tau \ln \pi_{k+1} =$$
$$r + \gamma P\big( \langle \pi_{k+1}, Q_k - \alpha\tau \ln \pi_k \rangle - \langle \pi_{k+1}, \alpha\tau(\ln \pi_{k+1} - \ln \pi_k) - (1-\alpha)\tau \ln \pi_{k+1} \rangle \big)$$
$$\Leftrightarrow Q'_{k+1} = r + \gamma P\big( \langle \pi_{k+1}, Q'_k \rangle - \alpha\tau D_{KL}(\pi_{k+1}||\pi_k) + (1-\alpha)\tau \mathcal{H}\left(\pi_{k+1}\right) \big) \tag{10}$$

where $Q'_{k+1} := Q_{k+1} - \alpha\tau \ln \pi_{k+1}$ is the generalized action value function.

The implementation of this idea uses the fact that $\alpha\tau \ln \pi_{k+1} = \alpha(Q_k - \mathcal{M}_\tau Q_k)$, where $\mathcal{M}_\tau Q_k := \frac{1}{Z_k} \left\langle \exp\left(\tau^{-1} Q_k\right), Q_k \right\rangle$, $Z_k = \left\langle \mathbf{1}, \exp\left(\tau^{-1} Q_k\right) \right\rangle$ is the Boltzmann softmax operator.[2] In the original work, computing this advantage term was found to be more stable than directly using the log of the policy. In our extension, we use the same form.

### 4.2 MVI($q$) For General $q$

The MVI(q) algorithm is a simple extension of MVI: it replaces the standard exponential in the definition of the advantage with the $q$-exponential. We can express this action gap as $Q_k - \mathcal{M}_{q,\tau}Q_k$, where $\mathcal{M}_{q,\tau}Q_k = \left\langle \exp_q\left(\frac{Q_k}{\tau} - \psi_q\left(\frac{Q_k}{\tau}\right)\right), Q_k \right\rangle$. When $q = 1$, it recovers $Q_k - \mathcal{M}_\tau Q_k$. We summarize this MVI($q$) algorithm in Algorithm B in the Appendix. When $q = 1$, we recover MVI. For $q = \infty$, we get that $\mathcal{M}_{\infty,\tau}Q_k$ is $\max_a Q_k(s, a)$—no regularization—and we recover advantage learning [Baird and Moore, 1999]. Similar to the original MVI algorithm, MVI($q$) enjoys tractable policy expression with $\pi_{k+1} \propto \exp_q\left(\tau^{-1} Q_k\right)$.

Unlike MVI, however, MVI($q$) no longer exactly implements the implicit regularization shown in Eq. (10). Below, we go through a similar derivation as MVI, show why there is an approximation and motivate why the above advantage term is a reasonable approximation. In addition to this reasoning, our primary motivation for this extension of MVI to use $q > 1$ was to inherit the same simple form as MVI as well as because empirically we found it to be effective.

Let us similarly define a generalized action value function $Q'_{k+1} = Q_{k+1} - \alpha\tau \ln_q \pi_{k+1}$. Using the relationship $\ln_q \pi_k = \ln_q \frac{\pi_k}{\pi_{k+1}} - \ln_q \frac{1}{\pi_{k+1}} - (1-q) \ln_q \pi_k \ln_q \frac{1}{\pi_{k+1}}$, we get

$$Q_{k+1} - \alpha\tau \ln_q \pi_{k+1} = r + \gamma P \langle \pi_{k+1}, Q_k + \alpha\tau \ln_q \pi_k - \alpha\tau \ln_q \pi_k + \tau S_q\left(\pi_{k+1}\right) \rangle$$
$$\Leftrightarrow Q'_{k+1} = r + \gamma P \langle \pi_{k+1}, Q'_k + \tau S_q\left(\pi_{k+1}\right) \rangle +$$
$$\gamma P \left\langle \pi_{k+1}, \alpha\tau \left( \ln_q \frac{\pi_k}{\pi_{k+1}} - \ln_q \frac{1}{\pi_{k+1}} - (1-q) \ln_q \frac{1}{\pi_{k+1}} \ln_q \pi_k \right) \right\rangle \tag{11}$$
$$= r + \gamma P \langle \pi_{k+1}, Q'_k + (1-\alpha)\tau S_q\left(\pi_{k+1}\right) \rangle - \gamma P \langle \pi_{k+1}, \alpha\tau D^q_{KL}(\pi_{k+1}||\pi_k) - \alpha\tau R_q(\pi_{k+1}, \pi_k) \rangle$$

---

[2]Using $\mathcal{M}_\tau Q$ is equivalent to the log-sum-exp operator up to a constant shift [Azar et al., 2012].

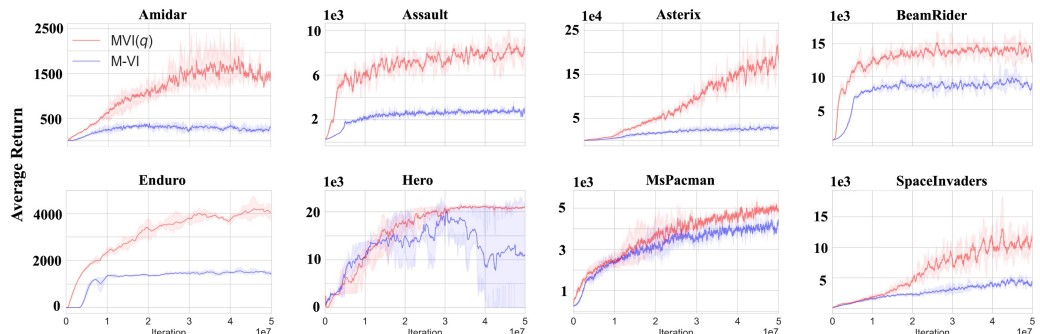

Figure 4: Learning curves of MVI($q$) and M-VI on the selected Atari games, averaged over 3 independent runs, with ribbon denoting the standard error. On some environments MVI($q$) significantly improve upon M-VI. Quantitative improvements over M-VI and Tsallis-VI are shown in Figures 5.

where we leveraged the fact that $-\alpha\tau\left\langle\pi_{k+1},\ln_q\frac{1}{\pi_{k+1}}\right\rangle = -\alpha\tau S_q(\pi_{k+1})$ and defined the residual term $R_q(\pi_{k+1},\pi_k) := (1-q)\ln_q\frac{1}{\pi_{k+1}}\ln_q\pi_k$. When $q=2$, it is expected that the residual term remains negligible, but can become larger as $q$ increases. We visualize the trend of the residual $R_q(\pi_{k+1},\pi_k)$ for $q=2,3,4,5$ on the `CartPole-v1` environment [Brockman et al., 2016] in Figure 3. Learning consists of $2.5\times 10^5$ steps, evaluated every 2500 steps (one iteration), averaged over 50 independent runs. It is visible that the magnitude of residual jumps from $q=4$ to 5, while $q=2$ remains negligible throughout.

A reasonable approximation, therefore, is to use $\ln_q\pi_{k+1}$ and omit this residual term. Even this approximation, however, has an issue. When the actions are in the support, $\ln_q$ is the unique inverse function of $\exp_q$ and $\ln_q\pi_{k+1}$ yields $\frac{Q_k}{\tau}-\psi_q\left(\frac{Q_k}{\tau}\right)$. However, for actions outside the support, we cannot get the inverse, because many inputs to $\exp_q$ can result in zero. We could still use $\frac{Q_k}{\tau}-\psi_q\left(\frac{Q_k}{\tau}\right)$ as a sensible choice, and it appropriately does use negative values for the Munchausen term for these zero-probability actions. Empirically, however, we found this to be less effective than using the action gap.

Though the action gap is yet another approximation, there are clear similarities between using $\frac{Q_k}{\tau}-\psi_q\left(\frac{Q_k}{\tau}\right)$ and the action gap $Q_k-\mathcal{M}_{q,\tau}Q_k$. The primary difference is in how the values are centered. We can see $\psi_q$ as using a uniform average value of the actions in the support, as characterized in Theorem 2. $\mathcal{M}_{q,\tau}Q_k$, on the other hand, is a weighted average of action-values.

We plot the difference between $Q_k-\mathcal{M}_{q,\tau}Q_k$ and $\ln_q\pi_{k+1}$ in Figure 3, again in Cartpole. The difference stabilizes around -0.5 for most of learning—in other words primarily just shifting by a constant—but in early learning $\ln_q\pi_{k+1}$ is larger, across all $q$. This difference in magnitude might explain why using the action gap results in more stable learning, though more investigation is needed to truly understand the difference. For the purposes of this initial work, we pursue the use of the action gap, both as itself a natural extension of the current implementation of MVI and from our own experiments suggesting improved stability with this form.

## 5   Experiments

In this section we investigate the utility of MVI($q$) in the Atari 2600 benchmark [Bellemare et al., 2013]. We test whether this result holds in more challenging environments. Specifically, we compare to standard MVI ($q=1$), which was already shown to have competitive performance on Atari [Vieillard et al., 2020b]. We restrict our attention to $q=2$, which was generally effective in other settings and also allows us to contrast to previous work [Lee et al., 2020] that only used entropy regularization with KL regularization. For MVI($q=2$), we take the exact same learning setup—hyperparameters and architecture—as MVI($q=1$) and simply modify the term added to the VI update, as in Algorithm 1.

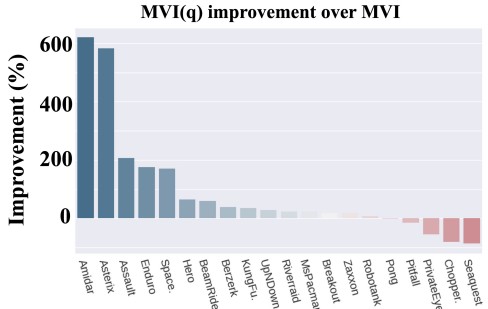 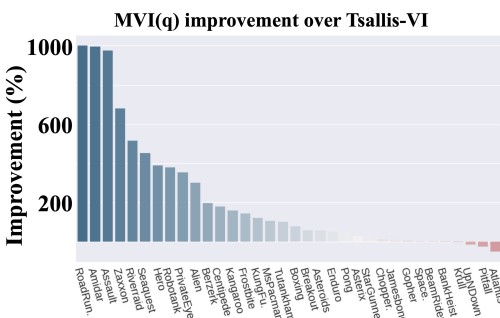

Figure 5: (Left) The percent improvement of MVI($q$) with $q = 2$ over standard MVI (where $q = 1$) on select Atari games. The improvement is computed by subtracting the scores from MVI($q$) and MVI and normalizing by the MVI scores. (Right) Improvement over Tsallis-VI on Atari environments, normalized with Tsallis-VI scores.

For the Atari games we implemented MVI($q$), Tsallis-VI and M-VI based on the Quantile Regression DQN [Dabney et al., 2018]. We leverage the optimized Stable-Baselines3 architecture [Raffin et al., 2021] for best performance and average over 3 independent runs following [Vieillard et al., 2020b], though we run 50 million frames instead of 200 million. From Figure 4 it is visible that MVI($q$) is stable with no wild variance shown, suggesting 3 seeds might be sufficient. We perform grid searches for the algorithmic hyperparameters on two environments `Asterix` and `Seaquest`: the latter environment is regarded as a hard exploration environment. MVI($q$) $\alpha : \{0.01, 0.1, 0.5, 0.9, 0.99\}$; $\tau : \{0.01, 0.1, 1.0, 10, 100\}$. Tsallis-VI $\tau : \{0.01, 0.1, 1.0, 10, 100\}$. For MVI we use the reported hyperparameters in [Vieillard et al., 2020b]. Hyperparameters can be seen from Table 2 and full results are provided in Appendix E.

## 5.1 Comparing MVI($q$) with $q = 1$ to $q = 2$

We provide the overall performance of MVI versus MVI($q = 2$) in Figure 5. Using $q = 2$ provides a large improvement in about 5 games, about double the performance in the next 5 games, comparable performance in the next 7 games and then slightly worse performance in 3 games (`PrivateEye`, `Chopper` and `Seaquest`). Both `PrivateEye` and `Seaquest` are considered harder exploration games, which might explain this discrepancy. The Tsallis policy with $q = 2$ reduces the support on actions, truncating some probabilities to zero. In general, with a higher $q$, the resulting policy is greedier, with $q = \infty$ corresponding to exactly the greedy policy. It is possible that for these harder exploration games, the higher stochasticity in the softmax policy from MVI whre $q = 1$ promoted more exploration. A natural next step is to consider incorporating more directed exploration approaches, into MVI($q = 2$), to benefit from the fact that lower-value actions are removed (avoiding taking poor actions) while exploring in a more directed way when needed.

We examine the learning curves for the games where MVI($q$) had the most significant improvement, in Figure 4. Particularly notable is how much more quickly MVI($q$) learned with $q = 2$, in addition to plateauing at a higher point. In `Hero`, MVI($q$) learned a stably across the runs, whereas standard MVI with $q = 1$ clearly has some failures.

These results are quite surprising. The algorithms are otherwise very similar, with the seemingly small change of using Munchausen term $Q_k(s, a) - \mathcal{M}_{q=2, \tau} Q_k$ instead of $Q_k(s, a) - \mathcal{M}_{q=1, \tau} Q_k$ and using the $q$-logarithm and $q$-exponential for the entropy regularization and policy parameterization. Previous work using $q = 2$ to get the sparsemax with entropy regularization generally harmed performance [Lee et al., 2018, 2020]. It seems that to get the benefits of the generalization to $q > 1$, the addition of the KL regularization might be key. We validate this in the next section.

## 5.2 The Importance of Including KL Regularization

In the policy evaluation step of Eq. (11), if we set $\alpha = 0$ then we recover Tsallis-VI which uses regularization $\Omega(\pi) = -S_q(\pi)$ in Eq. (1). In other words, we recover the algorithm that incorporates entropy regularization using the $q$-logarithm and the resulting sparsemax policy. Unlike MVI, Tsallis-

VI has not been comprehensively evaluated on Atari games, so we include results for the larger benchmark set comprising 35 Atari games. We plot the percentage improvement of MVI($q$) over Tsallis-VI in Figure 5.

The improvement from including the Munchausen term ($\alpha > 0$) is stark. For more than half of the games, MVI($q$) resulted in more than 100% improvement. For the remaining games it was comparable. For 10 games, it provided more than 400% improvement. Looking more specifically at which games there was notable improvement, it seems that exploration may again have played a role. MVI($q$) performs much better on `Seaquest` and `PrivateEye`. Both MVI($q$) and Tsallis-VI have policy parameterizations that truncate action support, setting probabilities to zero for some actions. The KL regularization term, however, likely slows this down. It is possible the Tsallis-VI is concentrating too quickly, resulting in insufficient exploration.

## 6    Conclusion and Discussion

We investigated the use of the more general $q$-logarithm for entropy regularization and KL regularization, instead of the standard logarithm ($q = 1$), which gave rise to Tsallis entropy and Tsallis KL regularization. We extended several results previously shown for $q = 1$, namely we proved (a) the form of the Tsallis policy can be expressed by $q$-exponential function; (b) Tsallis KL-regularized policies are weighted average of past action-values; (c) the convergence of value iteration for $q = 2$ and (d) a relationship between adding a $q$-logarithm of policy to the action-value update, to provide implicit Tsallis KL regularization and entropy regularization, generalizing the original Munchausen Value Iteration (MVI). We used these results to propose a generalization to MVI, which we call MVI($q$), because for $q = 1$ we exactly recover MVI. We showed empirically that the generalization to $q > 1$ can be beneficial, providing notable improvements in the Atari 2600 benchmark.

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

# A Basic facts of Tsallis KL divergence

We present some basic facts about $q$-logarithm and Tsallis KL divergence.

We begin by introducing the $2 - q$ duality for Tsallis statistics. Recall that the $q$-logarithm and Tsallis entropy defined in the main paper are:

$$\ln_q x = \frac{x^{1-q} - 1}{1 - q}, \quad S_q(x) = - \langle x^q, \ln_q x \rangle.$$

In the RL literature, another definition $q^* = 2 - q$ is more often used [Lee et al., 2020]. This is called the $2 - q$ duality [Naudts, 2002, Suyari and Tsukada, 2005], which refers to that the Tsallis entropy can be equivalently defined as:

$$\ln_{q^*} x = \frac{x^{q^* - 1} - 1}{q^* - 1}, \quad S_{q^*}(x) = - \langle x, \ln_{q^*} x \rangle,$$

By the duality we can show [Suyari and Tsukada, 2005, Eq.(12)]:

$$S_q(x) := - \left\langle x^q, \frac{x^{1-q} - 1}{1 - q} \right\rangle = \frac{\langle \mathbf{1}, x^q \rangle - 1}{1 - q} = \frac{\langle \mathbf{1}, x^{q^*} \rangle - 1}{1 - q^*} = - \left\langle x, \frac{x^{q^* - 1} - 1}{q^* - 1} \right\rangle =: S_{q^*}(x),$$

i.e. the duality between logarithms $\ln_{q^*} x$ and $\ln_q x$ allows us to define Tsallis entropy by an alternative notation $q^*$ that eventually reaches to the same functional form.

We now come to examine Tsallis KL divergence (or Tsallis relative entropy) defined in another form: $D_{KL}^q (\pi \,\|\, \mu) = \left\langle \pi, \ln_{q^*} \frac{\pi}{\mu} \right\rangle$ [Prehl et al., 2012]. In the main paper we used the definition $D_{KL}^q (\pi \,\|\, \mu) = \left\langle \pi, - \ln_q \frac{\mu}{\pi} \right\rangle$ [Furuichi et al., 2004]. We show they are equivalent by the same logic:

$$\left\langle \pi, - \ln_q \frac{\mu}{\pi} \right\rangle = \left\langle \pi, - \frac{\left( \frac{\mu}{\pi} \right)^{1-q} - 1}{1 - q} \right\rangle = \left\langle \pi, \frac{\left( \frac{\pi}{\mu} \right)^{q-1} - 1}{q - 1} \right\rangle = \left\langle \pi, \ln_{q^*} \frac{\pi}{\mu} \right\rangle. \tag{12}$$

The equivalence allows us to work with whichever of $\ln_q$ and $\ln_{q^*}$ that makes the proof easier to work out the following useful properties of Tsallis KL divergence:

— Nonnegativity $D_{KL}^q (\pi \,\|\, \mu) \geq 0$: since the function $- \ln_q \pi$ is convex, by Jensen's inequality

$$\left\langle \pi, - \ln_q \frac{\mu}{\pi} \right\rangle \geq - \ln_q \left\langle \pi, \frac{\mu}{\pi} \right\rangle = 0,$$

— Conditions of $D_{KL}^q (\pi \,\|\, \mu) = 0$: directly from the above, in Jensen's inequality the equality holds only when $\frac{\mu}{\pi} = 1$ almost everywhere, i.e. $D_{KL}^q (\pi \,\|\, \mu) = 0$ implies $\mu = \pi$ almost everywhere.

— Conditions of $D_{KL}^q (\pi \,\|\, \mu) = \infty$: To better align with the standard KL divergence, let us work with $\ln_{q^*}$, following [Cover and Thomas, 2006], let us define

$$0 \ln_{q^*} \frac{0}{0} = 0, \quad 0 \ln_{q^*} \frac{0}{\mu} = 0, \quad \pi \ln_{q^*} \frac{\pi}{0} = \infty.$$

We conclude that $D_{KL}^q (\pi \,\|\, \mu) = \infty$ whenever $\pi > 0$ and $\mu = 0$.

— Bounded entropy $\forall q$, $0 \leq S_q(\pi) \leq \ln_q |\mathcal{A}|$: let $\mu = \frac{1}{|\mathcal{A}|}$, by the nonnegativity of Tsallis KL divergence:

$$D_{KL}^q (\pi \,\|\, \mu) = \left\langle \pi, - \ln_q \frac{1}{(|\mathcal{A}| \cdot \pi)} \right\rangle = \left\langle \pi, \frac{(|\mathcal{A}| \cdot \pi)^{q-1} - 1}{q - 1} \right\rangle$$

$$= |\mathcal{A}|^{q-1} \left( \frac{\langle \mathbf{1}, \pi^q \rangle - 1}{q - 1} - \frac{\frac{1}{|\mathcal{A}|^{q-1}} - 1}{q - 1} \right) \geq 0.$$

Notice that $\frac{\langle \mathbf{1}, \pi^q \rangle - 1}{q - 1} = \left\langle \pi^q, \frac{1 - \pi^{1-q}}{1 - q} \right\rangle = \langle \pi, \ln_q \pi \rangle = -S_q(\pi)$ and $\frac{\frac{1}{|\mathcal{A}|^{q-1}} - 1}{q - 1} = \ln_q |\mathcal{A}|$, we conclude

$$S_q(\pi) \leq \ln_q |\mathcal{A}|.$$

# B    Proof of Theorem 1 and 2

We structure this section as the following three parts:

1. Tsallis entropy regularized policy has general expression for all $q$. Moreover, $q$ and $\tau$ are interchangeable for controlling the truncation (Theorem 1).
2. The policies can be expressed by $q$-exponential (Theorem 1).
3. We present a computable approximate threshold $\hat{\psi}_q$ (Theorem 2).

**General expression for Tsallis entropy regularized policy.**    The original definition of Tsallis entropy is $S_{q^*}(\pi(\cdot|s)) = \frac{p}{q^*-1}\left(1 - \sum_a \pi^{q^*}(a|s)\right), q^* \in \mathbb{R}, \ p \in \mathbb{R}_+$. Note that similar to Appendix A, we can choose whichever convenient of $q$ and $q^*$, since the domain of the entropic index is $\mathbb{R}$. To obtain the Tsallis entropy-regularized policies we follow [Chen et al., 2018]. The derivation begins with assuming an actor-critic framework where the policy network is parametrized by $w$. It is well-known that the parameters should be updated towards the direction specified by the policy gradient theorem:

$$\Delta w \propto \mathbb{E}_\pi \left[ Q_\pi \frac{\partial \ln \pi}{\partial w} + \tau \frac{\partial \mathcal{H}(\pi)}{\partial w} \right] - \sum_s \lambda(s) \frac{\partial \langle \mathbf{1}, \pi \rangle}{\partial w} =: f(w), \tag{13}$$

Recall that $\mathcal{H}(\pi)$ denotes the Shannon entropy and $\tau$ is the coefficient. $\lambda(s)$ are the Lagrange multipliers for the constraint $\langle \mathbf{1}, \pi \rangle = 1$. In the Tsallis entropy framework, we replace $\mathcal{H}(\pi)$ with $S_{q^*}(\pi)$. We can assume $p = \frac{1}{q^*}$ to ease derivation, which is the case for sparsemax.

We can now explicitly write the optimal condition for the policy network parameters:

$$\begin{aligned}
f(w) = 0 &= \mathbb{E}_\pi \left[ Q_\pi \frac{\partial \ln \pi}{\partial w} + \tau \frac{\partial S_{q^*}(\pi)}{\partial w} \right] - \sum_s \lambda(s) \frac{\partial \langle \mathbf{1}, \pi \rangle}{\partial w} \\
&= \mathbb{E}_\pi \left[ Q_\pi \frac{\partial \ln \pi}{\partial w} - \tau \frac{1}{q^*-1} \left\langle \mathbf{1}, \pi^{q^*} \frac{\partial \ln \pi}{\partial w} \right\rangle - \tilde{\psi}_q(s) \frac{\partial \ln \pi}{\partial w} \right] \\
&= \mathbb{E}_\pi \left[ \left( Q_\pi - \tau \frac{1}{q^*-1} \pi^{q^*-1} - \tilde{\psi}_q(s) \right) \frac{\partial \ln \pi}{\partial w} \right],
\end{aligned} \tag{14}$$

where we leveraged $\frac{\partial S_{q^*}(\pi)}{\partial w} = \frac{1}{q^*-1}\left\langle \mathbf{1}, \pi^{q^*} \frac{\partial \ln \pi}{\partial w} \right\rangle$ in the second step and absorbed terms into the expectation in the last step. $\tilde{\psi}_q(s)$ denotes the adjusted Lagrange multipliers by taking $\lambda(s)$ inside the expectation and modifying it according to the discounted stationary distribution.

Now it suffices to verify either $\frac{\partial \ln \pi}{\partial w} = 0$ or

$$Q_\pi(s,a) - \tau \frac{1}{q^*-1} \pi^{q^*-1}(a|s) - \tilde{\psi}_q(s) = 0$$

$$\Leftrightarrow \quad \pi^*(a|s) = \sqrt[q^*-1]{\left[ \frac{Q_\pi(s,a)}{\tau} - \frac{\tilde{\psi}_q(s)}{\tau} \right]_+ (q^*-1)},$$

$$\text{or} \quad \pi^*(a|s) = \sqrt[1-q]{\left[ \frac{Q_\pi(s,a)}{\tau} - \frac{\tilde{\psi}_q(s)}{\tau} \right]_+ (1-q)}, \tag{15}$$

where we changed the entropic index from $q^*$ to $q$. Clearly, the root does not affect truncation. Consider the pair $(q^* = 50, \tau)$, then the same truncation effect can be achieved by choosing $(q^* = 2, \frac{\tau}{50-1})$. The same goes for $q$. Therefore, we conclude that $q$ and $\tau$ are interchangeable for the truncation, and we should stick to the analytic choice $q^* = 2(q = 0)$.

**Tsallis policies can be expressed by $q$-exponential.**  Given Eq. (15), by adding and subtracting 1, we have:

$$\pi^*(a|s) = \sqrt[1-q]{\left[ 1 + (1-q)\left( \frac{Q_\pi(s,a)}{\tau} - \tilde{\psi}_q\left(\frac{Q_\pi(s,\cdot)}{\tau}\right) - \frac{1}{1-q} \right) \right]_+} = \exp_q\left( \frac{Q_\pi(s,a)}{\tau} - \hat{\psi}_q\left(\frac{Q_\pi(s,\cdot)}{\tau}\right) \right),$$

where we defined $\hat{\psi}_q = \tilde{\psi}_q + \frac{1}{1-q}$. Note that this expression is general for all $q$, but whether $\pi^*$ has closed-form expression depends on the solvability of $\tilde{\psi}_q$.

Let us consider the extreme case $q = \infty$. It is clear that $\lim_{q \to \infty} \frac{1}{1-q} \to 0$. Therefore, for any $x > 0$ we must have $x^{\frac{1}{1-q}} \to 1$; i.e., there is only one action with probability 1, with all others being 0. This conclusion agrees with the fact that $S_q(\pi) \to 0$ as $q \to \infty$: hence the regularized policy degenerates to $\arg\max$.

**A computable Normalization Function.** The constraint $\sum_{a \in K(s)} \pi^*(a|s) = 1$ is exploited to obtain the threshold $\psi$ for the sparsemax [Lee et al., 2018, Chow et al., 2018]. Unfortunately, this is only possible when the root vanishes, since otherwise the constraint yields a summation of radicals. Nonetheless, we can resort to first-order Taylor's expansion for deriving an approximate policy. Following [Chen et al., 2018], let us expand Eq. (15) by the first order Taylor's expansion $f(z) + f'(z)(x - z)$, where we let $z = 1$, $x = \left[ \frac{Q_\pi(s,a)}{\tau} - \tilde{\psi}_q \left( \frac{Q_\pi(s,\cdot)}{\tau} \right) \right]_+ (1 - q)$, $f(x) = x^{\frac{1}{1-q}}$, $f'(x) = \frac{1}{1-q} x^{\frac{q}{1-q}}$. So that the unnormalized approximate policy has

$$\tilde{\pi}^*(a|s) \approx f(z) + f'(z)(x - z)$$
$$= 1 + \frac{1}{1-q} \left( \left( \frac{Q_\pi(s,a)}{\tau} - \tilde{\psi}_q \left( \frac{Q_\pi(s,\cdot)}{\tau} \right) \right) (1 - q) - 1 \right). \tag{16}$$

Therefore it is clear as $q \to \infty$, $\tilde{\pi}^*(a|s) \to 1$. This concords well with the limit case where $\pi^*(a|s)$ degenerates to $\arg\max$. With Eq. (16), we can solve for the approximate normalization by the constraint $\sum_{a \in K(s)} \pi^*(a|s) = 1$:

$$1 = \sum_{a \in K(s)} \left[ 1 + \frac{1}{1-q} \left( \left( \frac{Q_\pi(s,a)}{\tau} - \tilde{\psi}_q \left( \frac{Q_\pi(s,\cdot)}{\tau} \right) \right) (1 - q) - 1 \right) \right]$$
$$= |K(s)| - \frac{1}{1-q} |K(s)| + \sum_{a \in K(s)} \left[ \frac{Q_\pi(s,a)}{\tau} - \tilde{\psi}_q \left( \frac{Q_\pi(s,\cdot)}{\tau} \right) \right]$$
$$\Leftrightarrow \tilde{\psi}_q \left( \frac{Q_\pi(s,\cdot)}{\tau} \right) = \frac{\sum_{a \in K(s)} \frac{Q_\pi(s,\cdot)}{\tau} - 1}{|K(s)|} + 1 - \frac{1}{1-q}.$$

In order for an action to be in $K(s)$, it has to satisfy $\frac{Q_\pi(s,\cdot)}{\tau} > \frac{\sum_{a \in K(s)} \frac{Q_\pi(s,\cdot)}{\tau} - 1}{|K(s)|} + 1 - \frac{1}{1-q}$. Therefore, the condition of $K(s)$ satisfies:

$$1 + i \frac{Q_\pi(s, a_{(i)})}{\tau} > \sum_{j=1}^{i} \frac{Q_\pi(s, a_{(j)})}{\tau} + i \left( 1 - \frac{1}{1-q} \right).$$

Therefore, we see the approximate threshold $\hat{\psi}_q = \tilde{\psi}_q + 1$. When $q = 0$ or $q^* = 2$, $\hat{\psi}_q$ recovers $\psi$ and hence $\tilde{\pi}^*$ recovers the exact sparsemax policy.

## C    Proof of convergence of $\Omega(\pi) = D_{KL}^q (\pi \,\|\, \cdot)$ when $q = 2$

Let us work with $\ln_{q^*}$ from Appendix A and define $\|\cdot\|_p$ as the $l_p$-norm. The convergence proof for $\Omega(\pi) = D_{KL}^q (\pi \,\|\, \cdot)$ when $q = 2$ comes from that $\Omega(\pi)$ is strongly convex in $\pi$:

$$\Omega(\pi) = D_{KL}^{q^*=2} (\pi || \cdot) = \left\langle \pi, \ln_2 \frac{\pi}{\cdot} \right\rangle = \left\langle \pi, \frac{\left( \frac{\pi}{\cdot} \right)^{2-1} - 1}{2 - 1} \right\rangle \propto \left\| \frac{\pi}{\cdot} \right\|_2^2 - 1. \tag{17}$$

Similarly, the negative Tsallis sparse entropy $-S_2(\pi)$ is also strongly convex. Then the propositions of [Geist et al., 2019] can be applied, which we restate in the following:

**Lemma 1** ([Geist et al., 2019]). *Define regularized value functions as:*

$$Q_{\pi,\Omega} = r + \gamma P V_{\pi,\Omega}, \qquad V_{\pi,\Omega} = \langle \pi, Q_{\pi,\Omega} \rangle - \Omega(\pi).$$

*If $\Omega(\pi)$ is strongly convex, let $\Omega^*(Q) = \max_\pi \langle \pi, Q \rangle - \Omega(\pi)$ denote the Legendre-Fenchel transform of $\Omega(\pi)$, then*

**Algorithm 1:** MVI($q$)

**Input:** number of iterations $T$, entropy coefficient $\tau$, TKL coefficient $\alpha$

Initialize $Q_0, \pi_0$ arbitrarily

Let $\{|\mathcal{A}|\} = \{1, 2, \ldots, |\mathcal{A}|\}$

**for** $k = 1, 2, \ldots, T$ **do**
   # Policy Improvement
   **for** $(s, a) \in (\mathcal{S}, \mathcal{A})$ **do**
      Sort $Q_k(s, a_{(1)}) > \cdots > Q_k(s, a_{(|\mathcal{A}|)})$
      Find $K(s) = \max \left\{ i \in \{|\mathcal{A}|\} \,\middle|\, 1 + i \frac{Q_k(s, a_{(i)})}{\tau} > \sum_{j=1}^{i} \frac{Q_k(s, a_{(j)})}{\tau} + i \left(1 - \frac{1}{1-q}\right) \right\}$
      Compute $\hat{\psi}_q \left( \frac{Q_k(s, \cdot)}{\tau} \right) = \frac{\sum_{a \in K(s)} \frac{Q_k(s,a)}{\tau} - 1}{|K(s)|} + 1$
      # Normalize when $q \neq 2$
      $\pi_{k+1}(a|s) \propto \exp_q \left( \frac{Q_k(s,a)}{\tau} - \hat{\psi}_q \left( \frac{Q_k(s,\cdot)}{\tau} \right) \right)$
   **end for**
   # Policy Evaluation
   **for** $(s, a, s') \in (\mathcal{S}, \mathcal{A})$ **do**
      $Q_{k+1}(s, a) =$
      $r(s, a) + \alpha\tau \left( Q_k(s, a) - \mathcal{M}_{q,\tau} Q_k(s) \right) + \gamma \sum_{b \in \mathcal{A}} \pi_{k+1}(b|s') \left( Q_k(s', b) - \tau \ln_q \pi_{k+1}(b|s') \right)$
   **end for**
**end for**

---

- $\nabla\Omega^*$ is Lipschitz and is the unique maximizer of $\arg\max_\pi \langle \pi, Q \rangle - \Omega(\pi)$.

- $T_{\pi,\Omega}$ is a $\gamma$-contraction in the supremum norm, i.e. $\|T_{\pi,\Omega} V_1 - T_{\pi,\Omega} V_2\|_\infty \leq \gamma \|V_1 - V_2\|_\infty$. Further, it has a unique fixed point $V_{\pi,\Omega}$.

- The policy $\pi_{*,\Omega} = \arg\max_\pi \langle \pi, Q_{*,\Omega} \rangle - \Omega(\pi)$ is the unique optimal regularized policy.

Note that in the main paper we dropped the subscript $\Omega$ for both the regularized optimal policy and action value function to lighten notations. It is now clear that Eq. (6) indeed converges for entropic indices that make $D_{KL}^q(\pi \,\|\, \cdot)$ strongly convex. But we mostly consider the case $q = 2$.

## D    Derivation of the Tsallis KL Policy

This section contains the proof for the Tsallis KL-regularized policy (7). Section D.1 shows that a Tsallis KL policy can also be expressed by a series of multiplications of $\exp_q (Q)$; while Section D.2 shows its more-than-averaging property.

### D.1    Tsallis KL Policies are Similar to KL

We extend the proof and use the same notations from [Lee et al., 2020, Appendix D] to derive the Tsallis KL regularized policy. Again let us work with $\ln_{q^*}$ from Appendix A. Define state visitation as $\rho_\pi(s) = \mathbb{E}_\pi \left[ \sum_{t=0}^\infty \mathbb{1}(s_t = s) \right]$ and state-action visitaion $\rho_\pi(s, a) = \mathbb{E}_\pi \left[ \sum_{t=0}^\infty \mathbb{1}(s_t = s, a_t = a) \right]$. The core of the proof resides in establishing the one-to-one correspondence between the policy and the induced state-action visitation $\rho_\pi$. For example, Tsallis entropy is written as

$$S_{q^*}(\pi) = S_{q^*}(\rho_\pi) = -\sum_{s,a} \rho_\pi(s, a) \ln_{q^*} \frac{\rho_\pi(s, a)}{\sum_a \rho_\pi(s, a)}.$$

This unique correspondence allows us to replace the optimization variable from $\pi$ to $\rho_\pi$. Indeed, one can always restore the policy by $\pi(a|s) := \frac{\rho_\pi(s,a)}{\sum_{a'} \rho_\pi(s,a')}$.

Let us write Tsallis KL divergence as $D_{KL}^{q^*}(\pi \,\|\, \mu) = D_{KL}^{q^*}(\rho \,\|\, \nu) = \sum_{s,a} \rho(s, a) \ln_{q^*} \frac{\rho(s,a) \sum_{a'} \nu(s,a')}{\nu(s,a) \sum_{a'} \rho(s,a')}$ by replacing the policies $\pi, \mu$ with their state-action visita-

tion $\rho, \nu$. One can then convert the Tsallis MDP problem into the following problem:

$$\max_\rho \sum_{s,a} \rho(s,a) \sum_{s'} r(s,a) P(s'|s,a) - D_{KL}^{q^*}(\rho \,\|\, \nu)$$

$$\text{subject to } \forall s,a, \quad \rho(s,a) > 0, \tag{18}$$

$$\sum_a \rho(s,a) = d(s) + \sum_{s',a'} P(s|s',a') \rho(s',a'),$$

where $d(s)$ is the initial state distribution. Eq. (18) is known as the Bellman Flow Constraints [Lee et al., 2020, Prop. 5] and is concave in $\rho$ since the first term is linear and the second term is concave in $\rho$. Then the primal and dual solutions satisfy KKT conditions sufficiently and necessarily. Following [Lee et al., 2020, Appendix D.2], we define the Lagrangian objective as

$$\mathcal{L} := \sum_{s,a} \rho(s,a) \sum_{s'} r(s,a) P(s'|s,a) - D_{KL}^{q^*}(\rho \,\|\, \nu) + \sum_{s,a} \lambda(s,a) \rho(s,a)$$

$$+ \sum_s \zeta(s) \left( d(s) + \sum_{s',a'} P(s|s',a') \rho(s',a') - \sum_a \rho(s,a) \right)$$

where $\lambda(s,a)$ and $\zeta(s)$ are dual variables for nonnegativity and Bellman flow constraints. The KKT conditions are:

$$\forall s,a, \quad \rho^*(s,a) \geq 0,$$

$$d(s) + \sum_{s',a'} P(s|s',a') \rho^*(s',a') - \sum_a \rho^*(s,a) = 0,$$

$$\lambda^*(s,a) \leq 0, \quad \lambda^*(s,a) \rho^*(s,a) = 0,$$

$$0 = \sum_{s'} r(s,a) P(s'|s,a) + \gamma \sum_{s'} \zeta^*(s') P(s'|s,a) - \zeta^*(s) + \lambda^*(s,a) - \frac{\partial D_{KL}^{q^*}(\rho^* \,\|\, \nu)}{\partial \rho(s,a)},$$

$$\text{where } - \frac{\partial D_{KL}^{q^*}(\rho^* \,\|\, \nu)}{\partial \rho(s,a)} = - \ln_{q^*} \frac{\rho^*(s,a) \sum_{a'} \nu(s,a')}{\nu(s,a) \sum_{a'} \rho^*(s,a')} - \left( \frac{\rho^*(s,a) \sum_{a'} \nu(s,a')}{\nu(s,a) \sum_{a'} \rho^*(s,a')} \right)^{q^*-1}$$

$$+ \sum_a \left( \frac{\rho^*(s,a)}{\sum_{a'} \rho^*(s,a')} \right)^{q^*} \left( \frac{\sum_{a'} \nu(s,a)}{\nu(s,a)} \right)^{q^*-1}.$$

The dual variable $\zeta^*(s)$ can be shown to equal to the optimal state value function $V^*(s)$ following [Lee et al., 2020], and $\lambda^*(s,a) = 0$ whenever $\rho^*(s,a) > 0$.

By noticing that $x^{q^*-1} = (q^* - 1) \ln_{q^*} x + 1$, we can show that $-\frac{\partial D_{KL}^{q^*}(\rho^* \,\|\, \nu)}{\partial \rho(s,a)} = -q^* \ln_{q^*} \frac{\rho^*(s,a) \sum_{a'} \nu(s,a')}{\nu(s,a) \sum_{a'} \rho^*(s,a')} - 1 + \sum_a \left( \frac{\rho^*(s,a)}{\sum_{a'} \rho^*(s,a')} \right)^{q^*} \left( \frac{\sum_{a'} \nu(s,a)}{\nu(s,a)} \right)^{q^*-1}$. Substituting $\zeta^*(s) = V^*(s)$, $\pi^*(a|s) = \frac{\rho^*(s,a)}{\sum_{a'} \rho^*(s,a)}$, $\mu^*(a|s) = \frac{\nu^*(s,a)}{\sum_{a'} \nu^*(s,a)}$ into the above KKT condition and leverage the equality $Q^*(s,a) = r(s,a) + \mathbb{E}_{s' \sim P}[\gamma \zeta^*(s')]$ we have:

$$Q^*(s,a) - V^*(s) - q^* \ln_{q^*} \frac{\pi(a|s)}{\mu(a|s)} - 1 + \sum_{a'} \pi(a|s) \left( \frac{\pi(a|s)}{\mu(a|s)} \right)^{q^*-1} = 0$$

$$\Leftrightarrow \pi^*(a|s) = \mu(a|s) \exp_{q^*} \left( \frac{Q^*(s,a)}{q^*} - \frac{V^*(s) + 1 - \sum_{a'} \pi(a|s) \left( \frac{\pi(a|s)}{\mu(a|s)} \right)^{q^*-1}}{q^*} \right).$$

By comparing it to the maximum Tsallis entropy policy [Lee et al., 2020, Eq.(49)] we see the only difference lies in the baseline term $\mu(a|s)^{-(q^*-1)}$, which is expected since we are exploiting Tsallis KL regularization. Let us define the normalization function as

$$\psi \left( \frac{Q^*(s,\cdot)}{q^*} \right) = \frac{V^*(s) + 1 - \sum_a \pi(a|s) \left( \frac{\pi(a|s)}{\mu(a|s)} \right)^{q^*-1}}{q^*},$$

then we can write the policy as

$$\pi^*(a|s) = \mu(a|s) \exp_{q^*} \left( \frac{Q^*(s,a)}{q^*} - \psi \left( \frac{Q^*(s,\cdot)}{q^*} \right) \right).$$

In a way similar to KL regularized policies, at $k+1$-th update, take $\pi^* = \pi_{k+1}, \mu = \pi_k$ and $Q^* = Q_k$, we write $\pi_{k+1} \propto \pi_k \exp_q Q_k$ since the normalization function does not depend on actions. We ignored the scaling constant $q^*$ and regularization coefficient. Hence one can now expand Tsallis KL policies as:

$$\pi_{k+1} \propto \pi_k \exp_{q^*}(Q_k) \propto \pi_{k-1} \exp_{q^*}(Q_{k-1}) \exp_{q^*}(Q_k) \propto \cdots \propto \exp_{q^*} Q_1 \cdots \exp_{q^*} Q_k,$$

which proved the first part of Eq. (7).

### D.2 Tsallis KL Policies Do More than Average

We now show the second part of Eq. (7), which stated that the Tsallis KL policies do more than average. This follows from the following lemma:

**Lemma 2** (Eq. (25) of [Yamano, 2002]).

$$\left( \exp_q x_1 \ldots \exp_q x_n \right)^{1-q} = \exp_q \left( \sum_{j=1}^{k} x_j \right)^{1-q} + \sum_{j=2}^{k} (1-q)^j \sum_{i_1=1<\cdots<i_j}^{k} x_{i_1} \cdots x_{i_j}. \tag{19}$$

However, the mismatch between the base $q$ and the exponent $1-q$ is inconvenient. We exploit the $q = 2 - q^*$ duality to show this property holds for $q^*$ as well:

$$\begin{aligned}
\left( \exp_{q^*} x \cdot \exp_{q^*} y \right)^{q^*-1} &= [1 + (q^*-1)x]_+ \cdot [1 + (q^*-1)y]_+ \\
&= \left[ 1 + (q^*-1)x + (q^*-1)y + (q^*-1)^2 xy \right]_+ \\
&= \exp_q(x+y)^{q^*-1} + (q^*-1)^2 xy.
\end{aligned}$$

Now since we proved the two-point property for $q^*$, by the same induction steps in [Yamano, 2002, Eq. (25)] we conclude the proof. The weighted average part Eq. (8) comes immediately from [Suyari et al., 2020, Eq.(18)].

## E   Implementation Details

We list the hyperparameters for Gym environments in Table 1. The epsilon threshold is fixed at $0.01$ from the beginning of learning. FC $n$ refers to the fully connected layer with $n$ activation units.

The Q-network uses 3 convolutional layers. The epsilon greedy threshold is initialized at 1.0 and gradually decays to 0.01 at the end of first 10% of learning. We run the algorithms with the swept hyperparameters for full $5 \times 10^7$ steps on the selected two Atari environments to pick the best hyperparameters.

We show in Figure 6 the performance of MVI($q$) on Cartpole-v1 and Acrobot-v1, and the full learning curves of MVI($q$) on the Atari games in Figure 7. Figures 8 and 9 show the full learning curves of Tsallis-VI.

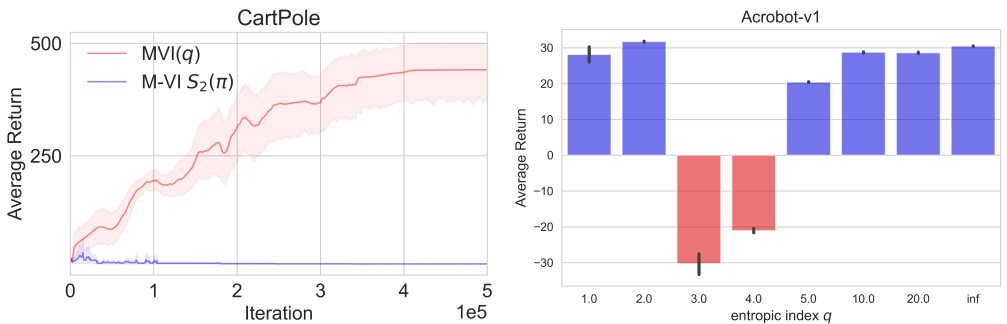

Figure 6: (Left) MVI($q$) and MVI with logarithm $\ln \pi$ simply replaced to $\ln_q \pi$ on Cartpole-v1, when $q = 2$. The results are averaged over 50 independent runs. The flat learning curve is due to the pseudo-additivity. (Right) MVI($q$) on Acrobot-v1 with different $q$ choices. Each $q$ is independently fine-tuned. The black bar stands for 95% confidence interval, averaged over 50 independent runs.

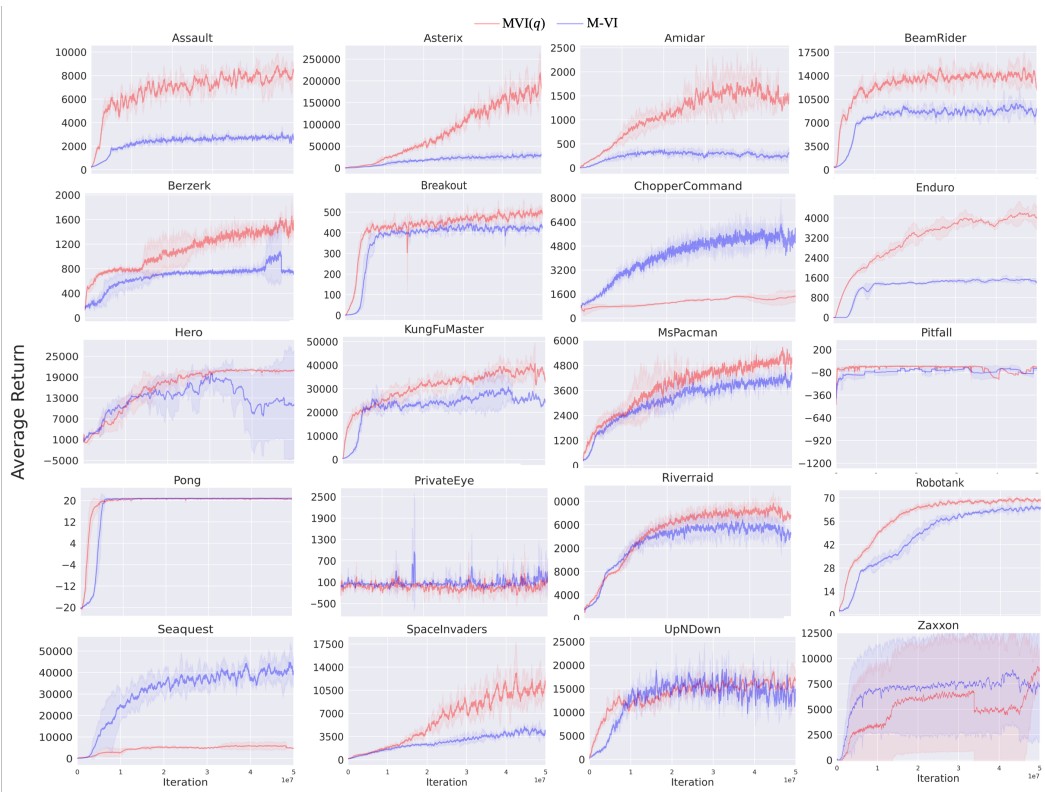

Figure 7: Learning curves of MVI($q$) and M-VI on the selected Atari games.

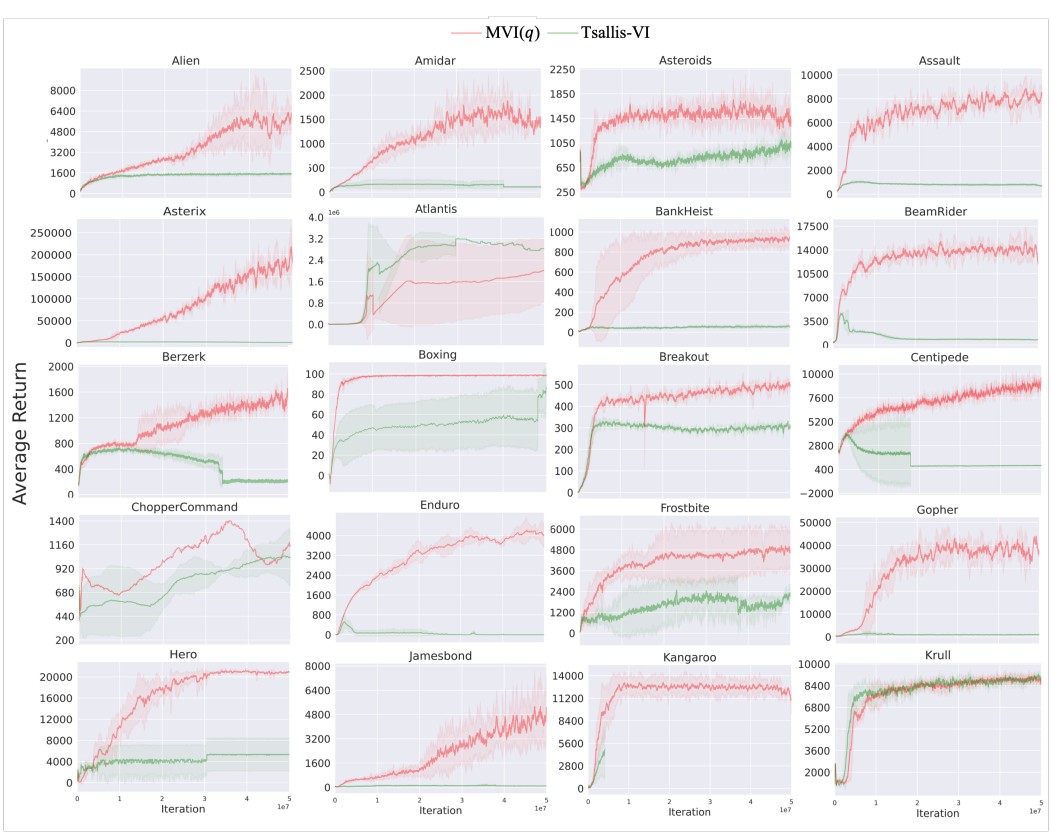

Figure 8: Learning curves of MVI($q$) and Tsallis-VI on the selected Atari games.

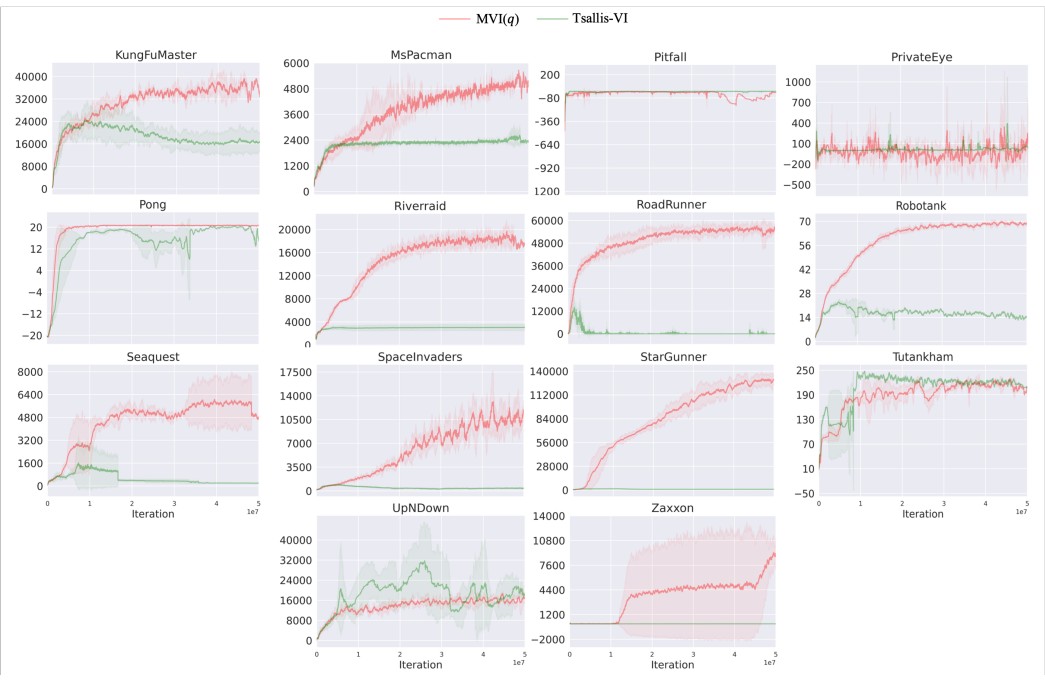

Figure 9: (cont'd) MVI($q$) and Tsallis-VI on the selected Atari games.

Table 1: Parameters used for Gym.

| Network Parameter | Value | Algorithm Parameter | Value |
|---|---|---|---|
| $T$ (total steps) | $5 \times 10^5$ | $\gamma$ (discount rate) | 0.99 |
| $C$ (interaction period) | 4 | $\epsilon$ (epsilon greedy threshold) | 0.01 |
| $|B|$ (buffer size) | $5 \times 10^4$ | $\tau$ (Tsallis entropy coefficient) | 0.03 |
| $B_t$ (batch size) | 128 | $\alpha$ (advantage coefficient) | 0.9 |
| $I$ (update period) | 100 (Car.) / 2500 (Acro.) | | |
| Q-network architecture | FC512 - FC512 | | |
| activation units | ReLU | | |
| optimizer | Adam | | |
| optimizer learning rate | $10^{-3}$ | | |

Table 2: Parameters used for Atari games.

| Network Parameter | Value | Algorithmic Parameter | Value |
|---|---|---|---|
| $T$ (total steps) | $5 \times 10^7$ | $\gamma$ (discount rate) | 0.99 |
| $C$ (interaction period) | 4 | $\tau_{\texttt{MVI}(q)}$ ( MVI($q$) entropy coefficient) | 10 |
| $|B|$ (buffer size) | $1 \times 10^6$ | $\alpha_{\texttt{MVI}(q)}$ ( MVI($q$) advantage coefficient) | 0.9 |
| $B_t$ (batch size) | 32 | $\tau_{\texttt{Tsallis}}$ (Tsallis-VI entropy coef.) | 10 |
| $I$ (update period) | 8000 | $\alpha_{\texttt{M-VI}}$ (M-VI advantage coefficient) | 0.9 |
| activation units | ReLU | $\tau_{\texttt{M-VI}}$ (M-VI entropy coefficient) | 0.03 |
| optimizer | Adam | $\epsilon$ (epsilon greedy threshold) | $1.0 \rightarrow 0.01|_{10\%}$ |
| optimizer learning rate | $10^{-4}$ | | |
| Q-network architecture | | | |
| $\quad$ Conv$_{8,8}^4$32 - Conv$_{4,4}^2$64 - Conv$_{3,3}^1$64 - FC512 - FC | | | |

