# OpenReview forum: "General Munchausen Reinforcement Learning with Tsallis Kullback-Leibler Divergence"
_NeurIPS.cc/2023/Conference — NeurIPS 2023 poster_

### Official Review · Reviewer_3pbN · 2023-06-20

**Soundness:** 2 fair
**Presentation:** 3 good
**Contribution:** 2 fair
**Rating:** 6
**Confidence:** 3

**Summary:**

This paper studies the Tsallis regularized MDPs, and proposes a practical algorithm for Tsallis KL divergence based on Munchausen RL. The experiments show that the resulting algorithm MVI($q$) performs notably better than its counterpart MVI.

**Strengths:**

1. The paper is well-written with sufficient introduction of background
2. Properties of Tsallis policies are sufficiently studied in the paper.

**Weaknesses:**

1. $\exp_q Q_2$ in Eq.(8) should be removed. In fact, since the permutation is changeable, no weighted average is performed in Eq.(8).
2. There is no clear theoretical justification nor intuition for why Tsallis regularization is beneficial for regularized value iteration

**Questions:**

1. Theorem 4 seems to suggest that the Tsallis KL regularized policy focuses more on the Q values at early iterations based on the second term in Eq.(7). Is this desirable?
2. Why would MVI($q$) provide notable gains while Tsallis value iteration provides no benefits?

**Limitations:**

The paper only considers $q>1$ for Tsallis regularization, but does not explore the case of $q<1$

---

> ### Author Rebuttal · Authors · 2023-08-10
>
> Addressing the reviewer’s comments on weakness:
>
> 1. We thank the reviewer for pointing out the typo $\exp_q Q_2$ in Eq. (8). It should be corrected as $\exp_q (\frac{Q_2}{1 + (1-q)Q_1})$. As the reviewer pointed out, permutation is changeable so there exists many way to expand the term $\exp_q (\sum_{i=1}^{k} Q_i)$, and our Eq. (8) serves mainly illustrating the point that TKL policy differs from  the uniform average of KL. We can also see the difference between TKL and KL policies by inspecting the second term in Eq. (7):  this action-value cross-product term further increases the probability for any actions that have had consistently larger values across iterations. This observation agrees with the mode-covering property of Tsallis KL.
>
>
>
>
> 2. It is true that we don’t have a theoretical justification for the benefits of Tsallis KL, but we would like to point out that this was the case for Shannon entropy and Tsallis entropy methods as well where only empirical evidence was provided. Our results do provide intuition and empirical evidence showing when Tsallis KL may be preferable.
>
>
> Specifically, at Line 84 we explained the superiority of KL regularization in terms of uniform average of history, and in Eq. (7) and (8) we explained how Tsallis KL regularization inherits the uniform average of history plus an additional cross-product term boosting actions that have had values consistently large across iterations. The average is not available to Tsallis regularized value iteration (Tsallis-VI). On the other hand, compared to MVI, Tsallis KL regularization provides an additional degree of freedom in truncating action support and therefore in better exploiting high probability actions, which is not available to Shannon entropy/KL divergence regularized methods like MVI.
>
>
> Answering the reviewer’s questions:
>
>
> 1.  We would like to point out that TKL regularized policy does not focus more on Q values at early iterations. Though policies at iteration $k+1$ can only make use of action values up to $k$, all action values are equally weighted. Instead, as we explained in the above, TKL policy does boost actions with consistently large values across iterations.
>
>
> 2. We hypothesize that it is the same reason that the KL divergence is usually better than using Shannon entropy: the KL results in a policy that averages over the history of action values, whereas Shannon entropy uses the most recent action values (see [Vieillard et al., 2020] *Leverage the average: an Analysis of KL Regularization in Reinforcement Learning*). MVI(q) has a parameter $\alpha$ that weights the KL regularization; for $\alpha = 0$, no KL regularization is used and we only have entropy regularization. In other words, for $\alpha = 0$, we do not get averaging over the history of action values. Our results for $q > 1$, therefore, parallel what is often observed for $q = 1$ (namely for KL divergence and Shannon entropy). Tsallis-VI uses Tsallis entropy only and corresponds to alpha=0. MVI(q) with $\alpha > 0$ averages over history of action value estimates to smooth out errors.
>
> Addressing the reviewer’s concern of not investigating $q<1$:
>
> We did not investigate $q<0$ since the function $-\ln_q x$ would be concave and no longer a member of $f$-divergence.
> We tested some values from $q \in (0,1)$, however, they all showed bad performance. Since Shannon entropy corresponds to $q=1$ (full support) and sparsemax entropy to $q=2$, no regularization to $q=\infty$, we have decided to focus on $q>1$.

---

> > ### Comment · Reviewer_3pbN · 2023-08-17
> >
> > Thank the authors for addressing my questions. I see that my question 1 relates to my misunderstanding of the meaning of $\sum_{i_1=1<\dots<i_j}^k$--I thought $i_1$ was always equal to $1$. I got the meaning after checking [T. Yamano. 2002](Some properties of q-logarithm and q-exponential functions in tsallis statistics), which wrote it clearer as $\sum_{1=i_1<\dots<i_j}^k$. Regarding the other questions addressed in response, I decide to raise my rating from 5 to 6.

---

### Official Review · Reviewer_6PUw · 2023-07-05

**Soundness:** 3 good
**Presentation:** 3 good
**Contribution:** 3 good
**Rating:** 6
**Confidence:** 3

**Summary:**

This paper introduces a principle way of generalizing KL-divergence regularized RL into Tsallis KL regularized RL. There have been studies that replaced Shannon entropy with Tsallis entropy to obtain sparsemax policies, but they had limited success. On the other hand, this paper extends Munchausen value iteration to get MVI(q). The extension is not straightforward due to the pseudo-additivity of $\ln_q$, but the paper manages to do it with approximation, and empirically show that the apprixmation error remains negligible for small $q$. In the experiments, the paper achieves large performance gain across 35 Atari games, where the authors conjecture that the improvement is due to the change in how MVI(q) do the exploration.

**Strengths:**

- Clear theoretical foundations for establishing the MVI(q) algorithm
- The paper is written clearly and easy to follow
- Proposed algorithm shows notable performance improvement over existing baselines

**Weaknesses:**

- The main characteristics/advantages of the proposed algorithm is less explained. While the authors assumed that the Tsallis policy can be better in exploitation allowing the policy to exploit high probability actions, It is also possible to control the regularization coefficient (i.e. temperature) to get similar effect, and it is hard to know the difference between these two options from the paper. The paper overall explains very well about the derivation of the algorithm, but it is hard to understand how it works and in what aspects it is better. The paper may benefit from including case scenarios on toy problems when MVI(q) works better than other algorithms.

**Questions:**

- The paper only explains in a way that MVI(q) does less exploration compared to MVI, and it better explores compared to Tsallis-VI. What happens if we control the regularization coefficient to negate thses effects (we can try adding shannon entropy regularization on top of Tsallis-VI if it concentrate to quickly)? If these effects are not able to be negated, in what aspects MVI(q) expores/exploits better than the other algorithms?
- How sensitive is MVI(q) to the regularization coefficient $\alpha$ compared to existing algorithms?

**Limitations:**

The authors have not addressed the limitations of their work.

---

> ### Author Rebuttal · Authors · 2023-08-10
>
> **Entropy**: Tsallis entropy truncates action support but Shannon entropy does not truncate. For softmax policies induced by Shannon entropy or KL divergence to ignore some actions, the temperature would have to be set to infinity, which is impossible in practice. On the other hand, the entropic index $q>1$ offers this flexibility. Moreover, for Tsallis-VI, adding Shannon entropy on top of Tsallis entropy would not only result in the loss of average of history but also sparsity, as can be seen from existing paper characterizing necessary conditions on the action sparsity, e.g., Theorem 2 of *A Regularized Approach to Sparse Optimal Policy in Reinforcement Learning* [Li et al., 2019]. In short, MVI(q) is capable of exploiting truncation of action support (controlled by $q$, not available to MVI ($q=1$)), as well as averaging the history of action values (controlled by $\alpha$, not available to Tsallis-VI).
>
> **Sensitivity**: We believe MVI(q) is similar in terms of sensitivity to $\alpha$. In Appendix Table 2 we provided hyperparameters of MVI(q) and MVI, which were respectively fine-tuned. They shared the same $\alpha=0.9$.
>
> As for the comment about not including limitations of our work. We do try to mention limitations throughout the work. In Section 3.3, we acknowledge that we as yet do not have a strong theoretical motivation for the policy form under Tsallis KL, and instead rely primarily on experiments to motivate it. The limitations of our approximation to obtain Tsallis KL regularization are discussed in line 230-239. However, it could be useful for the reader to have a paragraph dedicated to summarizing the limitations of the work, at the end of the paper. With the additional space for the final paper, we can add such a paragraph to the conclusion.

---

### Official Review · Reviewer_QmDX · 2023-07-10

**Soundness:** 4 excellent
**Presentation:** 3 good
**Contribution:** 3 good
**Rating:** 7
**Confidence:** 3

**Summary:**

The paper introduces the idea of Tsallis KL divergence for regularizing RL algorithms. They first introduce the Tsallis entropy based regularization formally whilst intuiting the how the q exponential and q logarithm, as defined by Tsallis 1998, have a truncation effect on the divergence. They also formalise this idea of truncation in Theorem 1. They also provide a provably tractable approximation of the *threshold function*. The authors then go on to very interestingly analyze Tsallis KL regularized policy and assert that "they average over history of value estimates". They finally introduce an algorithm similar to Munchausen Value Iteration (Vieillard et al., 2020b) which uses the Tsallis KL regularization instead of vanilla KL. Their empirical results show significant gains in a lot of environments.

**Strengths:**

The paper is largely well written. The manner in which the authors build on previous work in addition to explaining various concepts so well, section after section, is impressive. I liked how their theories and their empirical applications associated with Tsallis KL are explained via illustrations (figures 1,2,3). The empirical results suggest a strong improvement over MVI with numerous Atari environments. Figure 5 (left) shows that MVI(q) has significant improvements over MVI in various environments.

**Weaknesses:**

I see no major weaknesses. I have a couple of comments:
1. For section 3.3: is this average of histories a feature of Tsallis KL or is it to be expected of other KL regularization as well? Please let me know if this is mentioned somewhere (maybe line 183-184?). I would be interested in knowing how come this is unique for Tsallis KL case.
2. Section 4.2: you empirically choose to "omit the residual term" and judging from Section 5 this works well in practice. Despite this I would be interested in seeing the return on the cartpole environment as iterations increase. How is the the residual calculated or estimated for Figure 3? Please feel free to point me to the appendix or elsewhere in case I might have missed this.

Minor issues and fixes:
1. Line 77-78: citep for Sason and Verdu [2016]
2. Line 102: the use of the constant $p$ is not defined or explained while being introduced
3. How many seeds are the results of Figures 4 and 5 averaged over?
4. I would expect some discussion of the computational overhead especially in terms of GPU compute time MVI(q) adds to the algorithm.
5. For the sake of completeness, it would be beneficial to report all the hyper-params used by you even if you are porting them from past work. Additionally a detailed algorithm would also be a helpful addition in the Appendix.

**Questions:**

Please see the weaknesses section.

**Limitations:**

I do not see any major limitations.

---

> ### Author Rebuttal · Authors · 2023-08-10
>
> **Averaging**: Averaging is a feature of other KL regularization as well. The primary difference is the form of averaging. KL regularization induces a uniform average of the history, as can be seen from line 84. On the other hand, Tsallis KL inherits this uniform average plus an additional cross product term between action values, which boost the probability of actions that have had consistently large values across iterations.
>
>
> **Residual**:  The residual is given by line 230. We apologize for not including the details for computing the residual. To compute it, we store a copy of the target network for $Q_{k-1}$. Policies $\pi_{k+1}$ and $\pi_k$ are then respectively computed as shown in Algorithm 1. To prevent divide by zero issue, we clip the policy to the range $[0.01, 0.99]$.
> For the return on CartPole-v1, please refer to Figure 2 in the attached PDF. M-VI $S_2(\pi)$ denotes simply replacing the standard logarithm in MVI by the $q$-log, which performed poorly due to the pseudo-additivity of the $q$-log. We will include the residual computation procedure and Figure 2 to the final version of the paper.
>
>
> To answer the questions listed under minor issues and fixes:
>  - Atari results were averaged 3 seeds, and the error bars denote 95% confidence intervals. We apologize that we did not make this information clear, and will include it in the final version of the paper.
>  - Computational overhead of MVI(q) is approximately equal to MVI, with only an additional sorting for a list that is the length of  the number of actions, which is small for the environments we considered (please refer to the code or Algorithm 1 in page 16).
>  - A detailed algorithm for implementing MVI(q) is provided in page 16 of the appendix.

---

> > ### Comment · Reviewer_QmDX · 2023-08-16
> > **Response to the authors**
> >
> > Thank you for your response! I will stick to my score and I hope the authors can make the addition about the residuals in the revised version. Thank you for clarifying the averaging statement and for pointing me to the detailed algorithm.
> >
> > I would encourage the authors to increase the number of seeds for future versions to make the results more convincing.

---

### Official Review · Reviewer_FCv3 · 2023-07-26

**Soundness:** 2 fair
**Presentation:** 3 good
**Contribution:** 2 fair
**Rating:** 6
**Confidence:** 2

**Summary:**

Preface: this paper was assigned to me as an emergency review paper, so I had less time to do an in-depth review.

The paper tried to extend Munchausen Reinforcement learning by replacing the commonly used KL divergence with a more generalised form, i.e., Tsallis KL, and empirically demonstrated its benefits under various simulation experiments. The authors conduct theoretical proves to motivate their intuitive extension. Results showed that adding the Tsallis KL can bring impressive improvements over the simplest M-RL algorithm on game simulations.

=======post-rebuttal=========

Based on the clarifications and new experimental results, I raised my score to 6.

**Strengths:**

The paper is generally well-motivated and well-written at the beginning, with a clear introduction to new concepts such as the Tsallis KL Divergence and its intuitive effect visualisation as in Figure 2. It extended the standard KL-based MVI to a more generalised form MVI(q), supported by theoretical proof of the feasibility of using Tsallis KL instead of KL. Further discussion had been made to explore how Tsallis KL reweights the current policy, in the analogy of other well-recognised algorithms.

**Weaknesses:**

1.Contribution. The main contribution of this paper is the extension of Tsallis KL for M-RL and provided necessary math evidence of why can, which I sincerely appreciate. However, the authors fail to address why reweighting the policy distribution by the Tsallis q matters under the hood. From a critical point of view, this work extends an established work 3 years ago by replacing the policy regularisation function to a more variable form where a new hyper-parameter is introduced. Although the paper is self-contained, the contribution to RL community seems to be minor.

2.Potentially misleading and inadequate experiment result. The experiments only showed MVI(q) (i.e., the proposed method) with MVI and Tsallis-VI. The authors would be appreciated if they are able to include more baselines, especially those which considers policy regularisation, such as SAC, TRPO, etc. Moreover, the experiments were not sufficiently standardized. The authors may consider compare their methods with commonly acceptable benchmarks, such as Atari-57, or Mojuco. I also have some concern about Figure 5. The performance is obtained by computing “Improvement over Tsallis-VI on Atari environments, normalized with Tsallis-VI scores”, which, in an extreme case, look exaggerated when the Tsallis-VI scores are small enough. An easy fix is to compare MVI(q) to existing benchmark.

3.Lack of deeper explanation of experiments. It is yet clear to me how MVI(q) boosts performance. Figure 3 somehow tried to discuss it but still insufficient. The authors brought up many guesses in the paper, such as line 179 and line 272. However, they did not dig into any of those.


**Questions:**

Will the choice of q affect the performance significantly? Is it possible that some q choices are more suitable for some particular environments? If so, can you provide an empirical study of how to choose q adapting to the characteristic of task or environment?

---

> ### Author Rebuttal · Authors · 2023-08-10
>
> We appreciate you getting in an emergency review, and understand you did not have as much time. Your comments are nonetheless appreciated.
>
> We would like to address the overall goal and contribution of the paper. It is not yet certain if Tsallis KL regularization will prove to be an effective choice in RL, nor when it might be preferable to the standard KL regularization developed for max-ent RL. However, we cannot even begin to answer the utility of this regularization until we develop approaches to use Tsallis KL. The purpose of this paper is to provide such a strategy and to begin the investigation into its properties, knowing that it will take several papers to properly investigate this line of work. The original max-ent paper also did not fully answer the utility of entropy regularization; rather, this direction has become an important line of inquiry and we are starting to better understand when it is and is not useful.
>
> This paper 1) introduces Tsallis KL regularization to RL, a general class of regularizers that also includes regularizers like the alpha-divergence (see  e.g. Appendix A of [Li and Turner, 2016] or  [Wang et al., 2018, Belousov and Peters, 2019], mentioned in the  footnote on page 4.)  2) highlights theoretical properties of this regularizer including the form of the resulting policy, 3) provides some motivation for why this regularizer might be useful, 4) provides a practical, approximate implementation of the idea (MVI(q)) and 5) provides empirical evidence for the potential utility of generalizing q > 1. We are not setting out to get state-of-the-art, or show impressive performance on challenging benchmarks. We want to understand this regularizer. In that sense, we respectfully disagree that we needed to do benchmarks like Mujoco and compare to a different class of methods, like actor-critic methods (e.g, SAC). More on this below, explaining why we provide a normalized score to Tsallis-VI.
>
>
> Extending MVI to MVI(q) is by no means trivial. In fact, due to the properties of the Tsallis KL, it is downright difficult. We had to make approximations that took time to develop. Our extension does become the original MVI when $q=1$ (as we did intentionally), but MVI(q) for $q > 1$ is a totally new algorithm. Arguably, others might actually come up with better ways to approximate Tsallis KL regularization, because, as we discuss in the work, we had to make several approximation steps. We believe these to be reasonable, with some conceptual and empirical motivation, but they are approximations.
>
>
> We compare to Tsallis-VI because we are asking: given the same system/architecture, what is the impact of adding this new Tsallis KL regularization? As mentioned above, we are not trying to outperform SAC or be state-of-the-art. In that sense, the performance is not exaggerated, because we are not making a bold claim that it provides this level of improvement over all approaches. We are simply asking how much improvement is obtained when incorporating Tsallis KL instead of just Tsallis entropy regularization; the answer is that it can give significant improvement, for this agent.
>
>
> Finally, for your question about the choice of $q$, it can affect performance significantly. For many settings, $q = 1$ and $q = 2$ perform reasonably well, where $q = 1$ corresponds to the standard Shannon entropy/KL divergence and $q = 2$ is what was previously used for Tsallis entropy to get the sparsemax. One conclusion from this work is that shifting from $q = 1$ to $q = 2$ can often provide performance improvements, without even considering the other possible values of q. In other words, a reasonable choice for this generalized MVI is to use $q = 2$. However, we did find other values of q could also be effective, see Figure 1 in the uploaded PDF, on Acrobot-v1, with all $q$ independently fine-tuned. Looking at Eq. (3), different q not only control truncation but also the root, hence the smoothing effect. We do not yet know why certain $q$ might be better than others, and how it relates to properties of the environment; this is absolutely one of the important next steps for this work.

---

### Author Rebuttal · Authors · 2023-08-10

Included figures for rebuttal:
 - Figure 1: MVI(q) on Acrobot-v1 across different $q$
 - Figure 2: MVI(q) on CartPole-v1

---

### Decision · Program_Chairs · 2023-09-21

**Decision:**

Accept (poster)

**Comment:**

The paper investigates the effect of substituting the KL divergence, which is used as a regularizer, with a generalized KL divergence, called the Tsallis KL divergence. They perform empirical evaluation and present supporting theory.

The direction investigated by the paper is recognized to be interesting by the reviewers but to various extent.
Some limitations of the paper are be centered around limited as well as non-standard empirical evaluation.
Overall, however, the paper opens a new direction for research that is deemed interesting and which can lead to other works building on the present submission.